# A probe for NIR-II imaging and multimodal analysis of early Alzheimer's disease by targeting CTGF

Cao Lu[1], Cong Meng[1], Yuying Li[2], Jinling Yuan[1], Xiaojun Ren[1], Liang Gao[1], Dongdong Su[1], Kai Cao[1], Mengchao Cui[2], Qing Yuan[1]✉ & Xueyun Gao[1]✉

To date, earlier diagnosis of Alzheimer's disease (AD) is still challenging. Recent studies revealed the elevated expression of connective tissue growth factor (CTGF) in AD brain is an upstream regulator of amyloid-beta (Aβ) plaque, thus CTGF could be an earlier diagnostic biomarker of AD than Aβ plaque. Herein, we develop a peptide-coated gold nanocluster that specifically targets CTGF with high affinity (KD ~ 21.9 nM). The probe can well penetrate the blood-brain-barrier (BBB) of APP/PS1 transgenic mice at early-stage (earlier than 3-month-old) in vivo, allowing non-invasive NIR-II imaging of CTGF when there is no appearance of Aβ plaque deposition. Notably, this probe can also be applied to measuring CTGF on postmortem brain sections by multimodal analysis, including fluorescence imaging, peroxidase-like chromogenic imaging, and ICP-MS quantitation, which enables distinguishment between the brains of AD patients and healthy people. This probe possesses great potential for precise diagnosis of earlier AD before Aβ plaque formation.

Alzheimer's disease (AD) is the most prevalent neurodegenerative disease[1,2]. Accurate early diagnosis of AD permits timely intervention, thereby delays AD progression[3,4]. To date, imaging of Amyloid-β (Aβ) plaque in the brain is a hallmark of AD diagnosis[5,6]. However, disappointing results from putative disease-modifying therapies of Aβ have diverted researchers' attention to finding earlier biomarkers for AD[7]. Recent works revealed that elevated expression of connective tissue growth factor (CTGF) is strongly associated with the progression of clinical dementia and amyloid neuritic plaques (NP) in AD patients[8,9]. In AD postmortem brain, CTGF was upregulated in astrocytes and neurons that associated with senile plaques and the neuritic components of plaque, while there is no or low expression in normal brains[8,9]. In AD mice models, elevated CTGF expression in cortex and hippocampus was also found to be associated with increased amyloid plaque burden and the subsequent exacerbation of AD symptoms[9–11]. Mechanistic studies indicated that CTGF regulates amyloid precursor protein (APP) through upregulating γ-secretory enzyme, and ultimately promotes the elevation of Aβ plaque and its neuropathology[10–12]. A well-designed work indicated that in AD models, the specific elevated CTGF expression in the hippocampus or cortex appears at very early stages prior to Aβ accumulation[13]. Therefore, CTGF can act as a valuable hallmark for predicting the onset and progression of AD upstream of Aβ deposition[12,13]. Therefore, developing noninvasive probes for in vivo imaging of CTGF in the brain is a promising strategy for the earlier diagnosis of AD than Aβ plaque detection.

The ability to cross the blood–brain barrier (BBB) and the penetration depth of deep tissue imaging are two key factors that limit the diagnostic probe of AD[14,15]. Near-infrared-II (NIR-II) imaging has been considered as a promising noninvasive diagnostic method in vivo, for its advantages in deeper penetration depth, elevated imaging resolution and superior signal-to-noise ratio that owing to reduced photon scattering, light absorption and autofluorescence[16]. There are some reports for diagnosing AD by using Aβ-targeting NIR-II probes in animal model[17,18]. However, the NIR-II probe for CTGF imaging has not been reported yet.

[1]Center of Excellence for Environmental Safety and Biological Effects, Department of Chemistry, Beijing University of Technology, Beijing 100124, P. R. China. [2]Key Laboratory of Radiopharmaceuticals, Ministry of Education, College of Chemistry, Beijing Normal University, Beijing 100875, P. R. China. ✉e-mail: yuanqing@bjut.edu.cn; gaoxy@ihep.ac.cn

In this study, we synthesized a CTGF-targeting peptide-coated gold nanocluster that can emit both NIR-II and red fluorescence. The obtained probe (named DGC) is composed of 26 gold atoms and eight cyclic peptide ligands (named DAG). Notably, the probe is with more than 1000 times affinity to CTGF than free DAG peptides. Via the probe's intrinsic red fluorescence, we found that DGC specifically binds to CTGF in cells and brain sections of AD mouse in vitro. In addition, the precise gold composition of the DGC probe enables quantitative analysis of CTGC ex vivo by inductively coupled plasma mass spectrometry (ICP-MS). Also, the peroxidase-like activity of DGC can catalyze the 3, 3'-diaminobenzidine tertrahydrochloride (DAB) to form brown color products in situ, thus the expression of CTGF in cells and AD brain sections can be observed by naked eye directly. In AD mouse, the DGC probes can well home to the elevated CTGF in brain at earlier stages via NIR-II imaging, which precedes the appearance of Aβ plaques (Fig. 1). Finally, the probe can specifically label CTGF in brain sections of AD patients and detect CTGF by fluorescence imaging, ICP-MS quantification, and peroxidase-like catalyzed chromogenic imaging, which further demonstrated its potential in earlier AD diagnosis.

## Results

### The synthesis and characterization of DGC probe

The cyclic peptide modified-gold nanocluster was synthesized by a facile method (see supporting materials and methods). The sequence of the cyclic peptide is Cys-Asp-Ala-Gly-Arg-Lys-Gln-Lys-Cys (DAG). Its intrinsic sulfhydryl groups of the two adjacent cysteines were used to chelate Au nanoclusters via strong Au-S bonds (Fig. 2a). The acquired probe (DGC) is with good water solubility and the dynamic light scattering (DLS) measurement showed that the average hydrated size of DGC is ~2.85 nm (Fig. 2b). The gold nanocluster of DGC has an average size about 1.8 nm through high-resolution transmission electron microscopy (HRTEM) (Inset in Fig. 2b). The precise composition of DGC was determined by matrix-assisted laser desorption/ionization time-of-flight mass spectrometry (MALDI-TOF-MS). The strongest peak observed at 5634 $m/z$ was assigned to $Au_{26}S_{16}$ and the spaces between adjacent main peaks matched the molecular weight of single DAG peptide (Fig. 2c). Since one DAG peptide has two sulfur-containing cysteine residues, the precise molecular formula of DGC can be derived as $Au_{26}DAG_8$[19], e.g., one $Au_{26}$ gold nanocluster coated by eight DAG peptides (Fig. 2a). The aqueous solution of DGC emitted red fluorescence emission at 660 nm upon excitation by ultraviolet light, while DAG peptide alone showed no fluorescence (Fig. 2d and Fig. S1a). Notably, under the excitation of 808 nm, DGC has two obvious emission peaks at 923 and 1036 nm, respectively, and the 1036 nm emission can be used for in vivo NIR-II imaging (Fig. 2e and Fig. S1b). Moreover, peroxidase (POD)-like activities of DGC can be used to catalyze the oxidation of 3,3' 5,5'-tetramethylbenzidine sulfate (TMB) or 3,3'-diaminobenzidine tetrahydrochloride (DAB) to produce a visible color product in situ[20]. To test this, a TMB color assay was applied, in which different concentrations of DGC were added to the mixture of TMB and $H_2O_2$. The colorless solution gradually turned blue, and the intensity of blue is positively correlated with the concentration of DGC (Fig. 2f).

### DGC probe specifically targets CTGF in cells

CTGF is secreted by activated astrocytes and neurovascular endothelial cells in AD lesions[13]. To evaluate the CTGF-targeting ability of the DGC probe, we tested its affinity with CTGF by surface plasmon resonance (SPR) assay in vitro. The BIAcore assay showed that both the DGC probe and free DAG peptide were bound to CTGF (Fig. 3a). The dissociation equilibrium constants (KD) of DGC and free DAG were $2.19 \times 10^{-8}$ and $2.23 \times 10^{-5}$ M, respectively, indicating that the binding affinity of DGC to CTGF was about 1000 times higher than that of free DAG peptide. This result is consistent with previous reports that presenting multiple ligands on the probe surface can improve its affinity to the target[21,22]. The much stronger affinity of DGC may be due to the high density of eight DAG peptides on the surface of the probe. The specificity affinity between DGC and CTGF was further tested in three brain cell lines with different CTGF expression levels. Western blotting analysis in Fig. 3b showed that the expression of CTGF was highest in activated astrocytoma cells (U87MG), moderate in neuroblastoma cells (SH-SY5Y), and almost undetectable in normal astrocytes (CTX TNA2). DGC can effectively label CTGF in these cells and distinguish different CTGF expression levels in the three cell lines by fluorescence

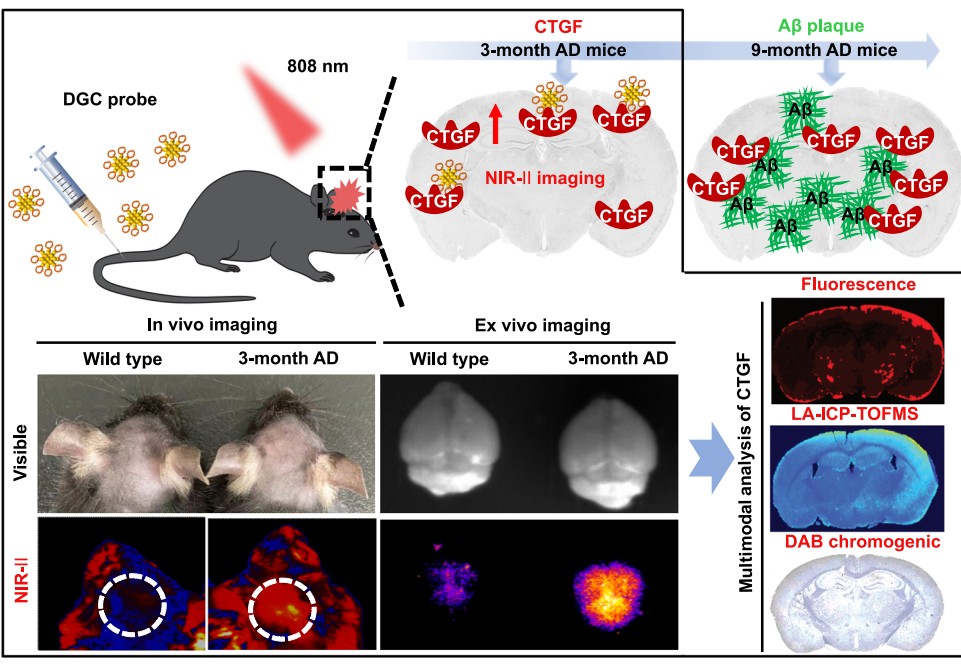

**Fig. 1 | Schematic of the DGC probe identifying early AD by NIR-II imaging and multimodal analysis.** The DGC probe can penetrate the blood–brain-barrier (BBB) of Alzheimer's disease (AD) mice at early-stage (earlier than 3-month-old) in vivo, allowing noninvasive NIR-II imaging of elevated connective tissue growth factor (CTGF) in AD brain prior to the appearance of amyloid-beta (Aβ) plaque deposition.

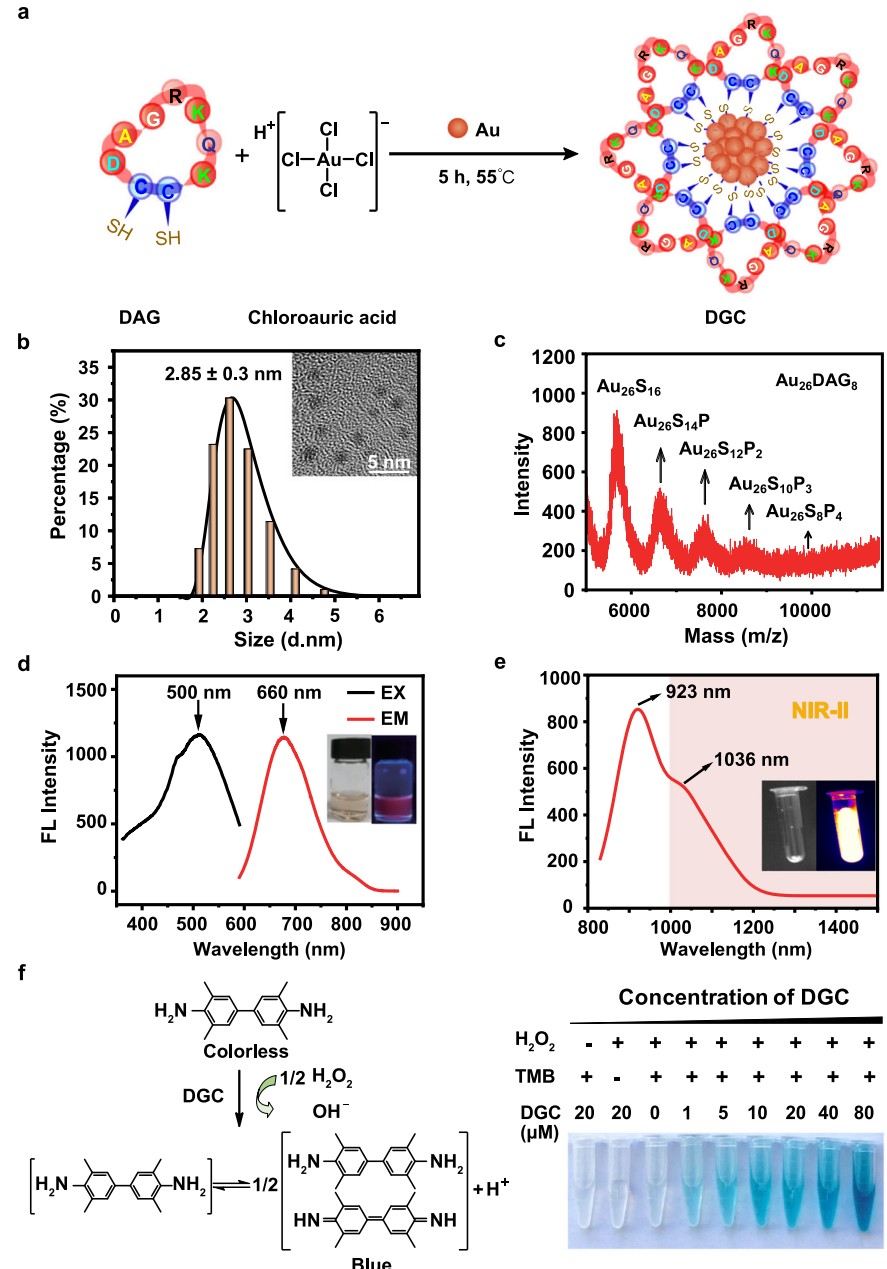

**Fig. 2 | Preparation and characterization of the DGC probe. a** Schematic illustration of DGC preparation. **b** Dynamic light scattering of DGC and Inset is the transmission electron microscopy (HRTEM) images of DGC. Scale bar = 5 nm. The experiment was repeated independently for more than three times with similar results. **c** MALDI-TOF-MS spectra of DGC in positive ion linear mode. The molecular formula of DGC is $Au_{26}DAG_8$. **d** Fluorescence excitation (black line, λ ex = 500 nm) and emission (red line, λ em = 660 nm) spectra of DGC. Inset are digital images of DGC under visible light (left) and 365 nm UV light (right), respectively. **e** NIR emission spectrum of DGC under 808 nm radiation (red line, λ em = 923 nm and 1036 nm). **f** Illustration of the peroxidase-like activity of DGC, serial concentration of DGC mixing with 3,3′ 5,5′-tetramethylbenzidine sulfate (TMB, 500 µM) and $H_2O_2$ (100 mM). Source data are provided as a Source Data file.

intensity (Fig. 3c and Fig. S2a) and inductively coupled plasma mass spectrometry (ICP-MS) analysis (Fig. S2b). Based on the peroxidase (POD)-like activity of DGC, DAB working solution was introduced into the DGC-labeled three cell lines and water-insoluble brown precipitate was subsequently formed in situ. The brown color intensity was also well correlated with the expression level of CTGF (fifth row in Fig. 3c and Fig. S2c). To further confirm the specificity of DGC targeting CTGF in cells, we stained U87MG cells with CTGF immunofluorescence antibody (green fluorescence) and DGC probe (red fluorescence) and observed significant colocalization (Fig. 3d). Furthermore, DGC probe cannot label CTGF in cells pre-blocked by free DAG peptides (Fig. 3d). Notably, DGC incubation did not cause any obvious cytotoxicity

to all three cell lines, indicating its biosafety and potential for in vivo application (Fig. S3). These results demonstrated the high specificity and sensitivity of DGC against CTGF-expressing cells in the brain, and DGC probes can be used for multimodal analysis of CTGF in cells via fluorescence imaging, ICP-MS quantification and chromogenic imaging.

## Multimodal analysis of CTGF in brain sections of AD mice

Based on the in vitro results in cell models, we wondered if DGC can identify CTGF in brain sections of AD mouse, especially at the early-stage when Aβ plaques have not appeared yet. APP/PS1 transgenic AD mice, a widely used mouse model of AD with age-related pathological

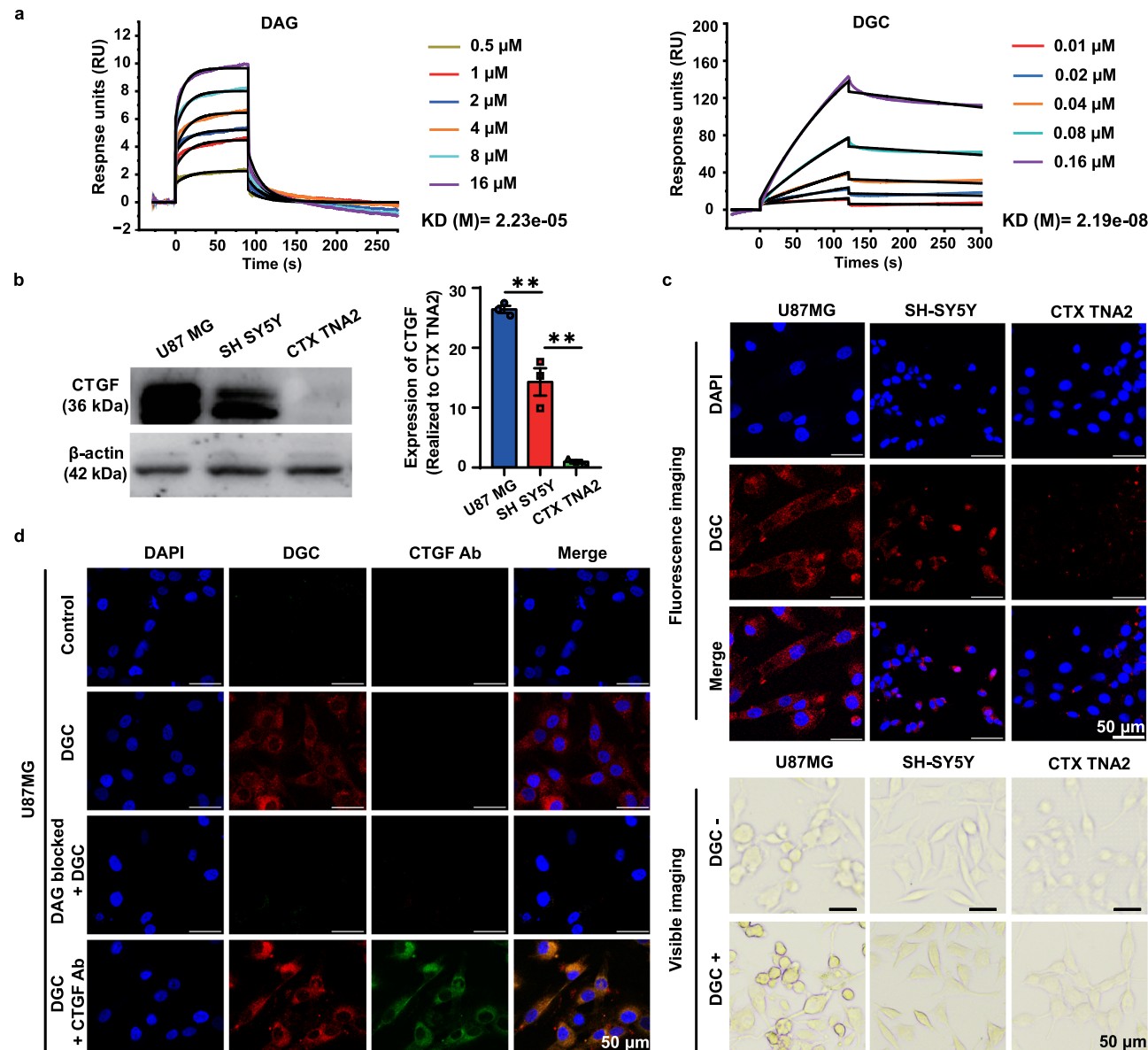

**Fig. 3 | The affinity and specificity of DGC for targeting CTGF. a** SPR data indicates the binding kinetics of free DAG (left) and DGC (right) with CTGF. The KD value of DAG is 22.3 μM and that of DGC is 21.9 nM. **b** Immunoblotting of CTGF was performed in U87MG, SH-SY5Y and CTX TNA2 cells. The CTGF expression levels are analyzed by imageJ software. Data were presented as mean ± SEM from three independent experiments ($n = 3$). **$P < 0.01$; Student's $t$-test. **c** Representative confocal laser scanning microscopy (CLSM) fluorescence (red) and visible chromogenic (brown) images of U87MG, SH-SY5Y, and CTX TNA2 cells after DGC labeling. **d** Representative CLSM images of U87MG cells labeled with DGC (red fluorescence) directly, or blocked with DAG peptide prior to DGC labeling (red fluorescence), or DGC (red fluorescence) co-staining with anti-CTGF antibody (green fluorescence). Data were from three independent experiments and one representative result is shown. Scale bar = 50 μm. Source data are provided as a Source Data file.

processes, and have obvious amyloid plaque formation starting at 8 months of age[23]. Thus, the CTGF expression in brain of APP/PS1 mice from early stages (1-, 2-, and 3-month-old) to establish stages (6- and 9-month-old) were evaluated by immunofluorescence staining with CTGF specific antibody in brain sections. The age-matched wild-type (WT) mice brain sections were used as controls. Results indicated that the expression of CTGF in cortex of APP/PS1 mice increased with the progression of the disease and was significantly higher than that in control WT mice at every age, even at 1-month old (Fig. 4a). To test the multimodal analysis ability of DGC probe in vitro, brain sections from early-stage (3-month-old) and Aβ plaque positive stage (9-month-old) of APP/PS1 mice were labeled by DGC, and the brain sections from age-matched WT mice were used as negative control. As shown in Fig. 4b, the overall expression level of CTGF in brain sections of AD mice can be

detected by DGC through red fluorescence imaging. The CTGF expression level in AD mice was significantly higher than that of the WT control. Moreover, the CTGF level was higher in the brain section of 9-month-old AD mice than that of 3-month-old, which was consistent with the pathological process of AD in APP/PS1 mice (Fig. 4b and Fig. S4a). The spatial distribution and expression level of CTGF on these brain sections was further analyzed by laser ablation inductively coupled plasma time-of-flight mass spectrometry (LA-ICP-TOF-MS), which can map the Au signal of DGC on brain sections by scanning method. The results indicated that elevated CTGF in the AD brain was mainly located in the cerebrocortex, which is consistent with the fluorescence intensity observation (Fig. 4b and Fig. S4b). When DAB working solution was added to the DGC-labeled AD brain sections, the POD-catalyzed brown product was deposited in situ, and its brown

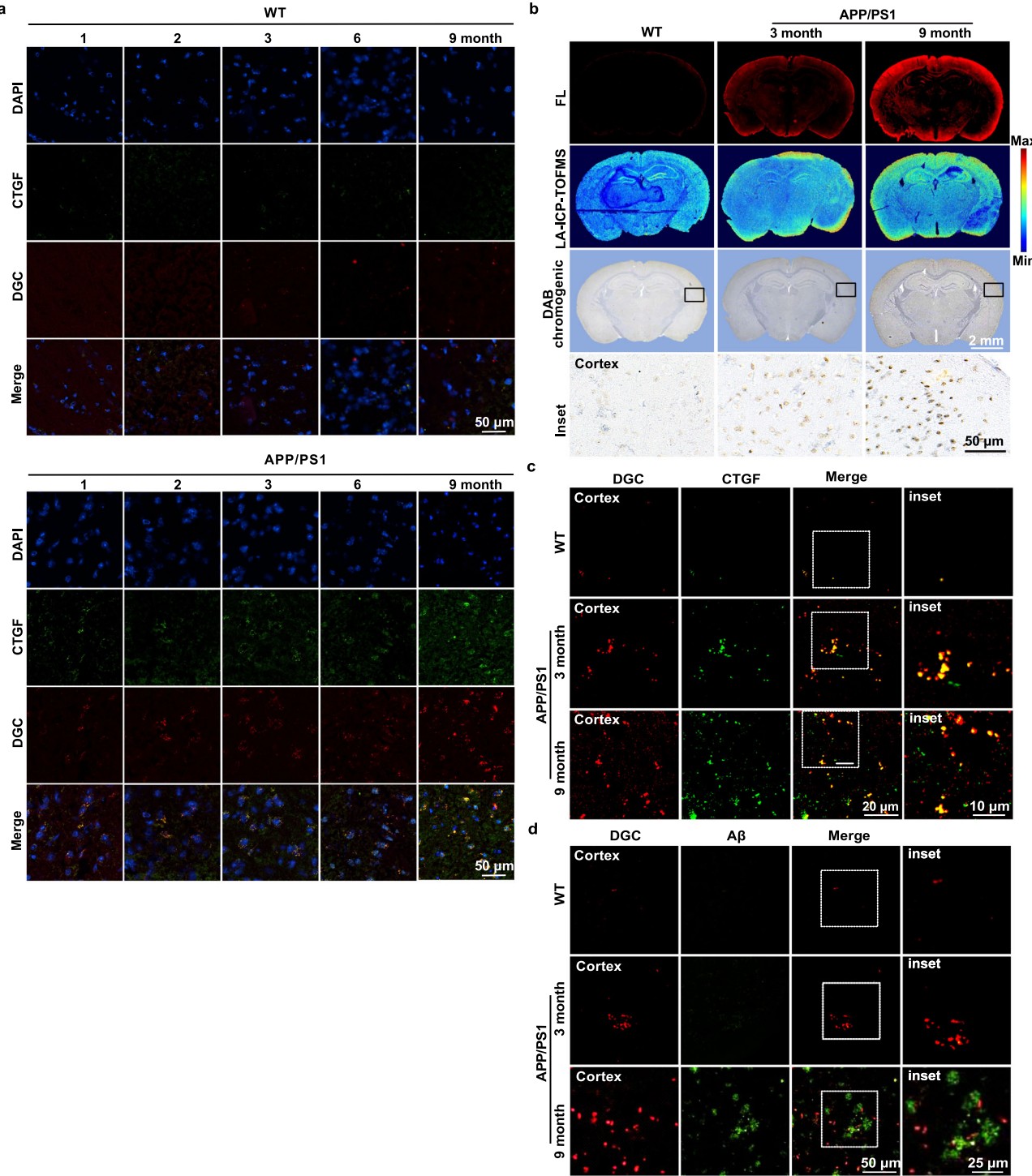

**Fig. 4 | DGC multimodally analyzes CTGF in early AD brain sections of APP/PS1 mice in vitro. a** Immunofluorescence staining of FITC-labeled CTGF antibody (green) and DGC (red) on APP/PS1 mouse brain sections and the age-matched wild-type (WT) brain sections from 1- to 9-month-old. The merge pictures show the colocalization of DGC and CTGF in brain sections. Scale bar = 50 µm. **b** In situ fluorescence imaging (red), Au elemental of LA-ICP-TOF-MS imaging, and 3, 3′-diaminobenzidine tetrahydrochloride (DAB) chromogenic (brown) imaging of CTGF in whole brain sections of WT mice (left), 3-month-old APP/PS1 mice (middle) and 9-month-old APP/PS1 mice (right). Inset pictures are enlarged areas of DAB chromogenic images. Scale bar = 50 µm. **c** CLSM fluorescence images of cortex region in brain sections of WT, 3-month-old APP/PS1 mice and 9-month-old APP/PS1 mice stained with DGC (red fluorescence) and FITC-labeled CTGF antibody (green fluorescence). Scale bar = 20 µm. Inset images show the colocalization of DGC and CTGF antibody. Scale bar = 10 µm. **d** CLSM fluorescence images of DGC (red fluorescence) and FITC-labeled-Aβ antibody (green fluorescence) staining show partial colocalization in cortex region in brain sections of WT, 3-month-old APP/PS1 mice and 9-month-old APP/PS1 mice. Scale bar = 50 µm. The inset pictures show the partial colocalization of DGC and Aβ antibody in 9-month-old APP/PS1 mice. Scale bar = 25 µm. Data were from three independent experiments.

intensity was also proportional to the expression of CTGF, which could distinguish the AD mouse brain from the WT mouse brain (Fig. 4b and Fig. S4c). Furthermore, the specificity of DGC against CTGF on brain sections was verified by colocalization with CTGF antibody. In Fig. 4c, the 3-month-old AD mice is with obvious higher level of CTGF than WT mice, and much more CTGF occurred in 9-month-old AD mice. Previous reports suggested that the elevated CTGF in AD brain is mainly produced by active astrocytes in the neurovascular unit, and we confirmed this conclusion by co-locating DGC with immunofluorescence of glial fibrillary acidic protein (GFAP, a marker of active astrocytes) in the APP/PS1 brain sections (Fig. S5). Aβ plaques formation in the brain was examined by immunofluorescence to verify the progression stage of AD in APP/PS1 mice. As shown in Fig. 4d, obvious Aβ plaques appeared in the brains of 9-month-old APP/PS1 mice, but no obvious plaques were detected in the brains of 3-month-old APP/PS1 mice and age-matched WT mice. In the brain section of 9-month-old APP/PS1 mice, some DGC-labeled CTGF is located in the vicinity of Aβ plaques (Fig. 4d), which is consistent with previous pathological reports[12,13]. Notably, obvious DGC-marked CTGF has been observed in the 3-month-old APP/PS1 brain sections without significant Aβ plaque deposition (Fig. 4c, d). All these data indicated that DGC can identify the CTGF in the early-stage AD brain before Aβ plaque formation (Fig. 4c, d).

### In vivo NIR-II imaging of CTGF in AD mice

Rare small molecule probes have been successfully used in NIR-II imaging of AD in vivo because the blood–brain barrier (BBB) hindrance[17,24]. To evaluate whether DGC could penetrate BBB and further target CTGF in the brains of AD mice, we test the probe in APP/PS1 mice from 1- to 9-month-old and the age-matched WT mice for in vivo and ex vivo NIR-II imaging of brains. First, a PAMPA assay mimicking BBB in vitro were performed to evaluate the potential penetration of DGC. The result showed the permeability Mean Pe of DGC is $2.15 \pm 0.44 \times 10^{-6}$ cm/s at 40 μM, indicating a high BBB permeability in vitro (Table S1). To optimize the time points for DGC NIR-II imaging in vivo. The stability and pharmacokinetics of DGC were determined in the mice model. Pharmacokinetic study in C57BL/6J mice suggested the elimination half-life ($t_{1/2z}$) of DGC in male and female mice was 21.38 h and 19.80 h, respectively (Table S2). NIR-II imaging and ICP-MS quantification of DGC in serum demonstrated its high stability during internal circulation (Fig. S6). In vivo NIR-II monitoring results showed that, in APP/PS1 mice, the brain NIR-II signals increased dramatically after tail vein injection of DGC and reached a peak at 4 h (Fig. S7). Therefore, this time point was used to evaluate the diagnostic capability of DGC for NIR-II imaging in AD mice and the age-matched WT mice. In vivo imaging showed that the brain fluorescence intensity of AD mice was higher than that of age-matched WT mice at all test ages, even as early as 1-month-old (Fig. 5a and S8). The results of ex vivo brain NIR-II imaging after cardiac perfusion of saline further verified the early diagnostic ability of DGC in APP/PS1 mice and also demonstrated the BBB penetration of DGC in vivo due to the cleaned cerebrovascular. The fluorescence intensity of the APP/PS1 brain was significantly stronger than that of the age-matched WT brain from 1- to 3-month-old (Fig. 5b). Ex vivo NIR-II imaging also confirmed the age-related elevation of CTGF expression in APP/PS1 mice brains, which was consistent with the immunofluorescence staining results of the brain sections in Fig. 4a (Fig. 5c, d).

### In vivo targeting of DGC to CTGF in AD mice brain

In order to clarify the NIR-II signal in AD brains when in vivo imaging is arisen from the DGC probe that bound to CTGF, we selected 3-month-old APP/PS1 mice representing early-stage AD and the age-matched WT mice as control to performed ex vivo multimodal analysis of the brain sections. As shown in Fig. 6a, the overall NIR-II fluorescence intensity in the brain of 3-month-old APP/PS1 mice is obviously higher than that of the WT control. To exclude the influence of vascular retention, we emptied the cerebral vessels by cardiac perfusion and performed ex vivo imaging of the brains. As shown in the fourth row of Fig. 6a, the brains with cleaned blood vessels still retained a strong NIR-II fluorescence signal, indicating that the DGC probe had crossed the BBB and reached the deep brain region. These results demonstrate that DGC can achieve noninvasive and real-time brain imaging in vivo. For brain sections made from AD mice after tail vein injection of DGC, the red fluorescence observation and LA-ICP-TOF-MS of DGC, as well as a POD-like catalysis test were performed. As shown in Fig. 6b, DGC maintains its stability in vivo and still enables multimodal analysis of CTGF in the brain. In Fig. 6c, the colocalization of DGC (red fluorescence) and CTGF (green fluorescence) on the ex vivo brain sections confirmed the specificity of DGC targeting CTGF in vivo. To further identify the cellular target of DGC in the brain of APP/PS1 mice, we carried out immunofluorescence staining on the ex vivo brain sections after tail vein injection of DGC where cell-specific fluorescent antibodies for microglia (Iba 1), neurons (NeuN), vascular endothelial cell (CD31) and reactive astrocytes (GFAP) were applied. In Fig. 6d, DAG colocalizes with GFAP-positive active astrocytes and CD31-positive microvessels, but not with neurons or microglia. Moreover, ex vivo NIR-II observation of major organs shows that DGC is mainly metabolized by the liver and excreted through the kidney and liver, which is suitable for in vivo imaging application (Fig. S9). The biosafety of DGC in C57BL/6J mice under the NIR-II imaging condition was evaluated by blood tests (blood cells and biochemical analysis) and organ pathological analysis (H&E staining). Results indicated that 20 mg/kg DGC did not induce any abnormalities in blood parameters and major organs, compared with the untreated normal mice, indicating good biosafety for in vivo diagnostic applications (Tables S3, S4 and Fig. S10).

### DGC probe specifically labels CTGF in brain sections from AD patients

While we have demonstrated that the DGC probe can detect CTGF in early AD mice, it is important to examine whether DGC also specifically marks CTGF in the brains of AD patients. To this end, we tested ex vivo targeting of DGC on postmortem brain sections from AD patients and health controls (HC) (Table S5, information on brain donors). As shown in Fig. 7a, the red fluorescent DGC well labeled the hippocampus and cortex of AD patient-derived brain sections, but not those from healthy volunteers. There were also abundant Aβ plaques (green fluorescence) in the same regions, colocalized with DGC. The colocalization of CTGF (green fluorescence) and DGC (red fluorescence) in the brain sections of AD patients was also observed, confirm the targeting specificity of DGC against CTGF in human brains (Fig. 7b). In contrast to the fluorescence-labeled antibody imaging of CTGF, DGC probe can also easily distinguish CTGF in AD brain sections through POD-catalyzed chromogenic imaging and ICP-MS quantification, respectively (Fig. 7c and Fig. S11). These results in human-derived AD samples suggested that DGC possessed the ability to distinguish CTGF in AD patients, thus raising possibilities for noninvasive diagnosis of AD patients in clinical settings.

### DGC identified AD patients in brain sections prior to Aβ accumulation

To support the sensitivity for the detection of early-onset AD patients, two brain sections from AD patients at a very early stage according to the Thal standard (Aβ accumulation stage I) and two healthy control samples were used to test DGC probe in vitro (Table S6, information on brain donors). As shown in Fig. 8a, b, the fluorescence signal of the DGC probe (red) in two early AD brain sections (Case 3 and Case 4) was obviously stronger and significantly different from that of the two healthy controls (Case 1 and Case 2). The quantitative results of ICP-MS

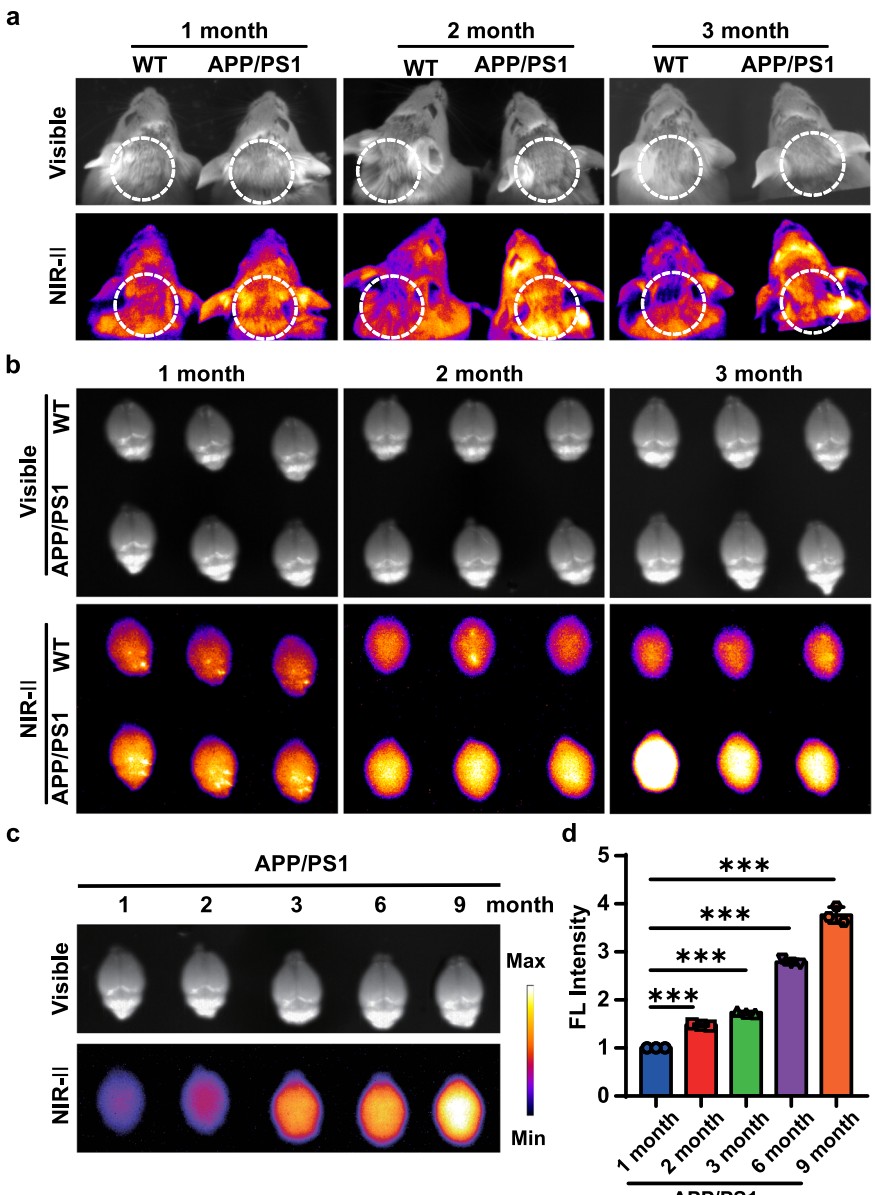

**Fig. 5 | In vivo and ex vivo NIR-II images of DGC in AD mice brains. a** The in vivo NIR-II images of APP/PS1 mice (right) and age-matched wild-type (WT) mice (left) from 1-month to 3-month-old after intravenous injection of DGC. **b** The ex vivo imaging of brains isolated from mice (cardiac perfusion with saline) after DGC injection for 4 h. **c** Ex vivo NIR-II imaging APP/PS1 mice brains aged from 1- to 9-month ($n = 3$). ($\lambda ex = 808$ nm). **d** Semi-quantitative analysis of the ex vivo NIR-II fluorescence (FL) intensity in APP/PS1 brains from 1-month to 9-month-old ($n = 3$). Data were presented as mean ± SD from three independent brains. ***$P < 0.001$ compared with the 1-month-old brain; Student's $t$-test. Source data are provided as a Source Data file.

also verified the significant difference (Fig. 8c). Moreover, the DGC probe (red) can co-locate well with CTGF immunofluorescence staining with antibody (green) on the AD brain sections (Fig. 8a). Considering the two AD patients are in Thal stage 1 without obvious Aβ deposition, these results indicated that DGC probe can identify AD brain prior to obvious Aβ accumulation, with a potential for early diagnosis of AD patient.

## Discussion

Growing clinical evidence suggested earlier intervention may bring greater benefits to AD patients[3,4]. However, inadequate early diagnosis is a global problem in AD management[25]. Aβ plaques and later neurofibrillary tangles (NFTs) have long been recognized as two hallmarks for AD diagnosis and therapy[5,6]. But the disappointing results from clinical trials targeting these two targets have diverted researchers' attention to earlier diagnosis and intervention prior to Aβ

accumulation[7]. Recent works suggested that the elevated CTGF in the brain could be an emerging predictor for the early diagnosis of AD[8–13]. Initially, strong CTGF immunoreactivity was observed in the vicinity of Aβ plaques and NFTs in AD postmortem brains, which mainly appeared in the hippocampus, entorhinal cortex, and temporal cortex and increased with the progression of the disease[8,9]. In AD mice models, elevated CTGF expression in the brain was also found to be associated with the increased amyloid plaque burden and the subsequent exacerbated AD symptoms[9–13]. When the level of CTGF was reduced in APP/PS1 transgenic AD mice by tauroursodeoxycholic acid (TUDCA) supplemented, hippocampal and prefrontal amyloid deposition were decreased, and the spatial, recognition and contextual memory defects were also prevented[10,12]. These results demonstrated that CTGF is an upstream regulator of Aβ, and its elevated expression in AD brain is significantly precedes the appearance of Aβ plaques, which could be used for early AD diagnosis[13].

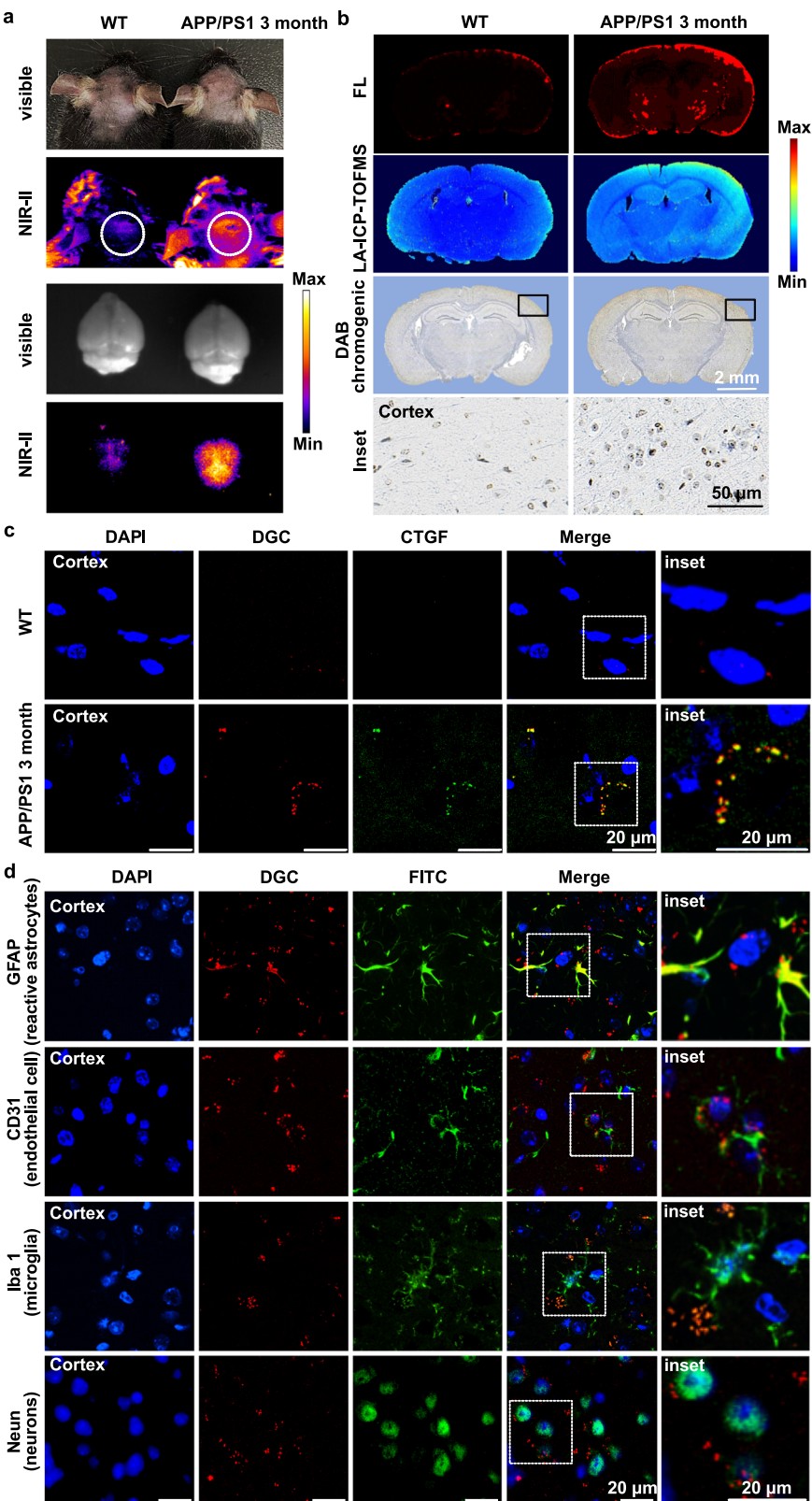

NIR-II (1000–1700 nm) probes showed more attractive than conventional visible fluorescent probes for in vivo diagnosis, due to noninvasive, high signal-to-noise ratio, deeper penetration depth, and elevated imaging resolution[16,26]. However, although some probes targeting Aβ oligomers have been attempted, there are still no NIR-II probes available for AD early diagnosis to date[18]. Therefore, NIR-II probes targeting CTGF may offer a promising opportunity.

The blood–brain barrier (BBB) prevents most molecules from entering the central nervous system, hindering brain imaging applications of many NIR-II probes[24]. Recently, atomically precise ultrasmall gold nanoclusters (AuNCs) have emerged as potential in vivo probes for bioimaging, due to their emitting of NIR-II photoluminescence (PL) and good biocompatibility[27–29]. In particular, the good water solubility and excellent BBB penetration ability makes them promising for

**Fig. 6 | Specific DGC marking and imaging of CTGF in AD brain of 3-month-old APP/PS1 mice in vivo. a** The in vivo NIR-II fluorescence images of wild-type (WT) mice (left) and 3-month-old APP/PS1 mice (right) after intravenous injection of DGC. And the ex vivo imaging of brains isolated from mice (cardiac perfusion with saline) after DGC injection ($\lambda$ ex = 808 nm). The experiment was repeated in three independent pair of mice with similar results. **b** In situ fluorescence (red) imaging, LA-ICP-TOF-MS imaging, and DAB chromogenic (brown) imaging of CTGF in brain sections prepared from the ex vivo NIR-II analyzed brains. Inset pictures are enlarged areas of DAB chromogenic imaging. Scale bar = 50 µm. Each experiment was repeated independently for three times with similar results. **c** The brain sections prepared from the brains after ex vivo NIR-II imaging, sections were stained with FITC-labeled CTGF antibody (green fluorescence). Scale bar = 20 µm. The inset showed the colocalization of DGC (red fluorescence) and CTGF antibody (green fluorescence). Scale bar = 20 µm. The experiment was repeated independently in three pairs of mice with similar results. **d** Representative images of DGC (red fluorescence) location in cells of brain sections. GFAP marked reactive astrocytes, CD31 marked vascular endothelial cells, Neun marked neurons, and Iba 1 marked microglia cells (all green fluorescence). Scale bar = 20 µm. The inset showed the colocalization of DGC (red fluorescence) and CTGF (green fluorescence) expressed by reactive astrocytes and vascular endothelial cells rather than microglia cells and neurons. Each experiment was repeated independently three times with similar results.

application in brain imaging[30,31]. As expected, we synthesized a CTGF-targeting peptide-coated AuNC for earlier AD detection even when there is no obvious Aβ plaque deposition. The probe provides NIR-II imaging and multimodal measurement of CTGF in the brains of early-stage AD mice when the CTGF expression level is significantly higher than that of wild-type control and no significant Aβ plaque appears. This probe can also label the higher-expressed CTGF in brain sections of AD patients and allows CTGF detection by fluorescence imaging, chromogenic imaging, and quantity mass analysis. These data inspire researchers to apply this probe for earlier AD diagnosis even without obvious Aβ plaque occurrence.

Understanding the exact origins of NIR-II PL of AuNCs is of great significance for the development of highly effective NIR-II probes[29]. However, the mechanism is very complicated and still under debate[32]. So far, only $Au_{25}$ clusters have been systematically studied for the correlation of PL origins to their structures, through experimental characterizations and theoretical calculations[33]. By examining different surface ligands, it was shown that the NIR emission of the $Au_{25}$ cluster likely originated from the $Au_{13}$ icosahedral core state[33]. It was found that the NIR-II PL of AuNCs can be enhanced by increasing the rigidity of surface ligands[34], and the intrinsic structural rigidity also strongly affects the NIR PL quantum efficiency of thiolated AuNCs[35]. Recent studies indicated the great significance of the central Au atom and the surface "lock rings" as well the surface "lock atoms" in enhancing the NIR-II PL of AuNCs by suppressing the nonradiative decay[36,37]. In summary, the NIR-II PL emission of AuNCs is related to the Au core and the surface ligands as well as the charge transfer states, which leads to the complexity of their fluorescence origin[38].

Although sufficient to produce bright signals in AD mice, the PL quantum yield of DGC was not very high. Many studies have focused on enhancing the NIR-II PL of AuNCs, but it remains a challenging task[29]. Currently, atom doping and ligand engineering have been shown to be two major promising strategies for enhancing the NIR-II PL of AuNCs[26,39–41]. In particular, the NIR-II PL brightness and quantum yield (QY) of AuNCs can be significantly improved by single hetero-metallic atom doping, for example, $Au_{25}Cd_1$NCs exhibited about 56 times higher NIR-II PL intensity than $Au_{25}$NCs[42], which may be the direction of further optimization of DGC probes.

## Methods

### Ethical statement
The research complies with all relevant ethical regulations. All mice were purchased from Beijing HFK Bioscience Co., Ltd and the post-mortem brain sections of AD patients and health controls were obtained from Chinese Brain Bank Center (CBBC) and Human Brain Bank, Central South University Xiangya School of Medicine. The protocol used in all animal experiments and human-derived samples detection was approved by the Ethics Committee of Beijing University of Technology, China (Protocol # HS202202009).

### Chemicals, cells, and animals
The peptide (CDAGRKQKC, named DAG) was synthesized by a solid-phase method (Beijing Scilight Biotechnology LLC. Purity: 95%). Sodium hydroxide (NaOH), nitric acid ($HNO_3$, MOS grade), hydrochloric acid (HCl, MOS grade), and hydrogen peroxide ($H_2O_2$, MOS grade) were obtained from Beijing Chemical Reagent Co., China. Hydrogen tetrachloroaurate (III) ($HAuCl_4 \cdot 3H_2O$) was purchased from Sinopharm Chemical Reagent Co., Ltd. Human astrocytoma cell line (U87MG), neuroblastoma cell line (SH-SY5Y) and astrocytes line (CTX TNA2) were purchased from the Cancer Institute and Hospital, Chinese Academy of Medical Sciences. The double transgenic (APP/PS1) AD model mice and wild-type C57BL/6 J mice were purchased from Beijing HFK Bioscience Co., Ltd., and maintained under standard condition. Brain samples of AD patients are obtained from the Chinese Brain Bank Center (CBBC) and Human Brain Bank, Central South University Xiangya School of Medicine. Cell culture medium DMEM/High Glucose, phosphate buffer solution (PBS), fetal bovine serum, and trypsin EDTA were acquired from Gibco, USA. Primary antibodies were purchased from Proteintech, Abcam, BioLegend, and Bioss. A colorimetric Cell Counting Kit-8 (CCK-8) was purchased from Dojindo Laboratory, Kumamoto, Japan. Cell lysis buffer for immunoblotting, DAPI staining solution, IgG (H + L) (HRP-labeled Goat Anti-Rabbit IgG (H + L)), Blocking Buffer for Western Blot, DAB Horseradish Peroxidase Color Development Kit and a BCA Protein Assay Kit and 4% Paraformaldehyde Fix Solution were obtained from the Beyotime Institute of Biotechnology (Shanghai, China). All other materials were commercially available and used as received unless otherwise mentioned. The water with a resistivity of 18.2 MΩ.cm, used throughout the experiment, was purified with a Milli-Q system from Millipore Co.

### Synthesis of DGC probe
Briefly, 20 µL $HAuCl_4$ (25 mM aqueous solution) was slowly added to the DAG peptide solution (1 mM, 500 µL) with stirring. Subsequently, 10 µL freshly prepared sodium citrate (40 mM) was added to the mixture. Next, 100 µL sodium hydroxide solution (0.5 mM) was added to the reaction system drop by drop. The whole process takes place at 55 °C and seals away from light for 5 h to obtain DGC. Before use, all glassware should be soaked in aqua regia (HCl: $HNO_3$, volume ratio = 3:1) and then rinsed with ultrapure water. The assembled DGC were purified and concentrated using an ultrafiltration tube (Millipore, MWCO: 3 KDa) by adding $H_2O$ to cut off peptide and free ions, leaving the solution at a pH of 9.0. The purified sample was sealed and protected from light at 4 °C for later studies.

### Characterization of DGC probe
The luminescence of obtained DGC was observed in the dark by using a portable ultraviolet analyzer (ZF-5, Nanjing, China). The fluorescence spectrophotometer (Shimadzu RF-5301, Kyoto, Japan) and NIR spectrograph FLS1000 was used to acquire the fluorescence and NIR spectra of the as-synthesized DGC. The mean hydrated particle size and size distribution of DGC were further documented by a phase analysis light-scattering technique (Zetasizer Nano, Malvern). The synthetic DGC was also observed by using a High-Resolution Transmission Electron Microscope (HRTEM, FEI Talos F200X-G2) at the accelerated voltage of 200 kV. Matrix-assisted laser desorption/ionization time-of

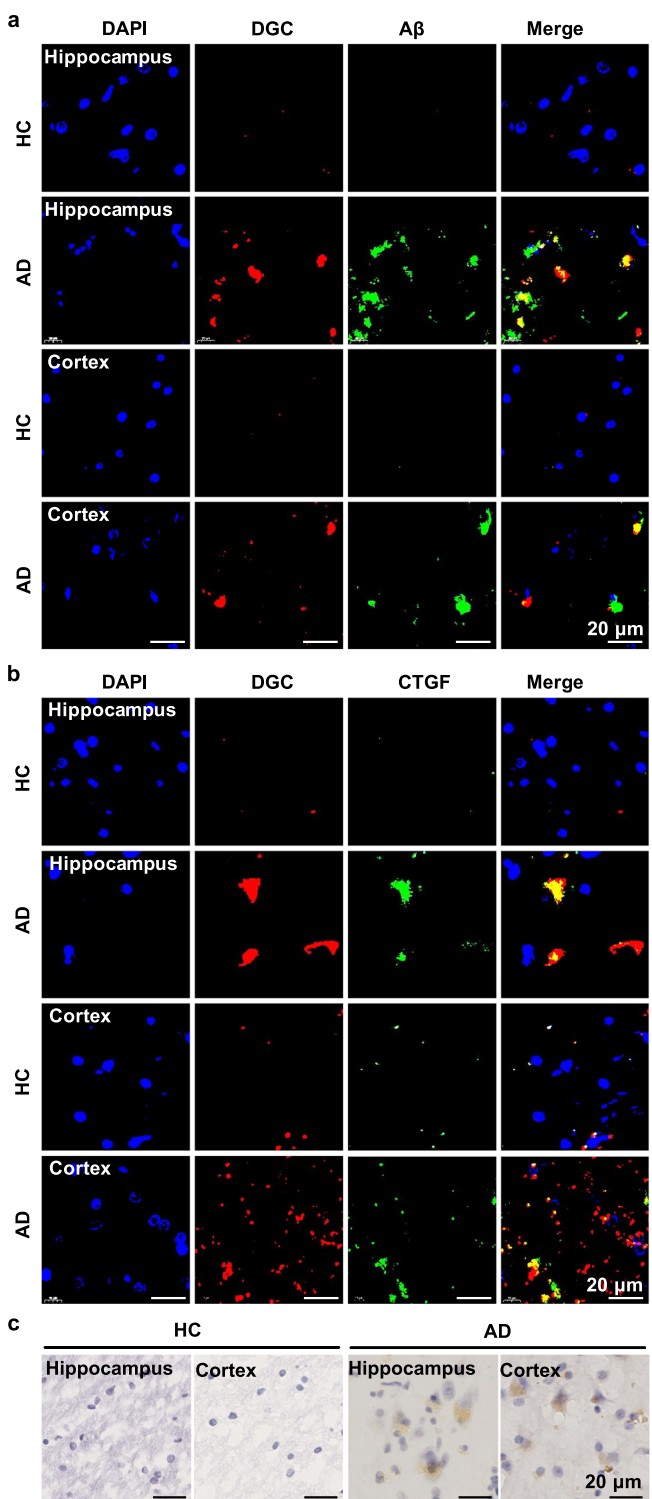

**Fig. 7 | Specific DGC marking and imaging of CTGF in AD patient-derived brain sections and healthy control (HC) brain sections. a** The fluorescence imaging of hippocampal and cortex regions in brain sections from AD patient and HC after staining with DGC (red fluorescence) and FITC-labeled-Aβ antibody (green fluorescence). The merge pictures show the partial colocalization of DGC and Aβ. Scale bar = 20 μm. **b** The fluorescence imaging of hippocampal and cortex regions in brain sections from AD patient and HC after staining with DGC (red fluorescence) and FITC-labeled CTGF antibody (green fluorescence). The merge pictures indicate the colocalization of DGC and CTGF. Scale bar = 20 μm. **c** Representative chromogenic imaging (brown) of hippocampal and cortex regions in brain sections from AD patient and HC after staining with DGC and DAB working solution. Scale bar = 20 μm. Each experiment was repeated independently in two AD patient-derived brain sections and two HC brain sections with similar results.

## Affinity between DGC and CTGF protein by SPR analysis

Real-time binding and kinetic analyses by surface plasmon resonance (SPR) were performed on a BIAcore 8 K$^+$ instrument (Pharmacia Biosensor AB). The recombinant Human CTGF protein (Abcam, Cat#ab283457) was immobilized on a CM5 chip by using an amine coupling kit, and the remaining coupling sites were blocked with ethanolamine. The eluent contained PBS and 0.005% Tween 20. Binding was evaluated over a range of DAG (0.5–16 μM) or DGC (0.01–0.16 μM) concentrations at 25 °C. Kinetic parameters were further determined with Biacore Insight Evaluation 3.0 software.

## CTGF expression in cells via Western blot

We evaluated the expression level of CTGF in human astrocytoma cells (U87MG), neuroblastoma cells (SH-SY5Y), and astrocytes (CTX TNA2) by a traditional immunoblotting method. U87MG, SH-SY5Y, and CTX TNA2 cells were plated in a six-well plate at a density of $5 \times 10^5$ cells per well. Subsequently, the cells were collected when harvested and washed with cold PBS. Then, the total protein was extracted through RIPA lysis buffer. Protein concentration was tested by the BCA protein assay kit. The protein samples were added in 10% sodium dodecyl sulfate (SDS-PAGE) gel prepared beforehand and transferred to a (polyvinylidene fluoride) (PVDF) membrane. The membrane was incubated with the targeted primary CTGF antibodies (dilution 1:1000, Abacm, Cat#ab6992) at 4 °C overnight after being blocked by blocking buffer for western blot. Afterward, the membrane was immunized with HRP-labeled secondary antibodies for 1 h at room temperature. After being washed with TBST, chemiluminescent HRP substrates (Millipore) were added to the membrane and a gel-imaging system (Tanon, 5200 Multi, China) was applied to capture the chemiluminescence. The gray value of the CTGF band was analyzed using ImageJ software. Uncropped blots were presented in the Source Data file.

## Cytotoxicity of DGC

U87MG, SH-SY5Y, and CTX TNA2 cells were seeded into 96-well plates at a density of $5 \times 10^3$ cells per well and were incubated at 37 °C for 24 h. Then, the cells were treated with 5, 10, 20, 50, 100, and 500 μM DGC for 24 h. Next, 10 μL of a colorimetric Cell Counting Kit (CCK-8) reagent diluted with 100 μL medium was added to each well, and incubated for 30 min, The absorbance at 450 nm was measured using a microplate reader (Molecular Devices, San Jose, CA, USA). The relative viability was normalized to control cells. The experiment was performed in triplicates.

## Evaluating the specificity of DGC to CTGF in cells by fluorescence imaging

The U87MG, SH-SY5Y, and CTX TNA2 cells in confocal dishes were fixed with 4% paraformaldehyde solution (PFA) for 30 min. Next, the cells were washed with cold PBS three times, and 3% BSA in PBS solution was introduced to the cells to block the nonspecific recognition sites. Then, the cells were incubated with 55 μM DGC for 60 min at room temperature. The control cells were only incubated with a

flight mass spectrometry (MALDI-TOF-MS, ultrafleXtreme, Bruker, Germany) was used to characterize the molecular composition of the as-synthesized cluster.

## Peroxidase-like activity of DGC in vitro

Under the catalysis of HRP or other peroxidases, 3, 3', 5, 5'- tetramethylbenzidine (TMB) solution can produced soluble blue color product. Here, the concentration-dependent peroxidase-like activity of DGC assays was carried out with a series of differing DGC concentrations (0, 1, 5, 10, 20, 40, and 80 μM) at 100 mM $H_2O_2$ and 500 μM TMB solution system at pH 7.0.

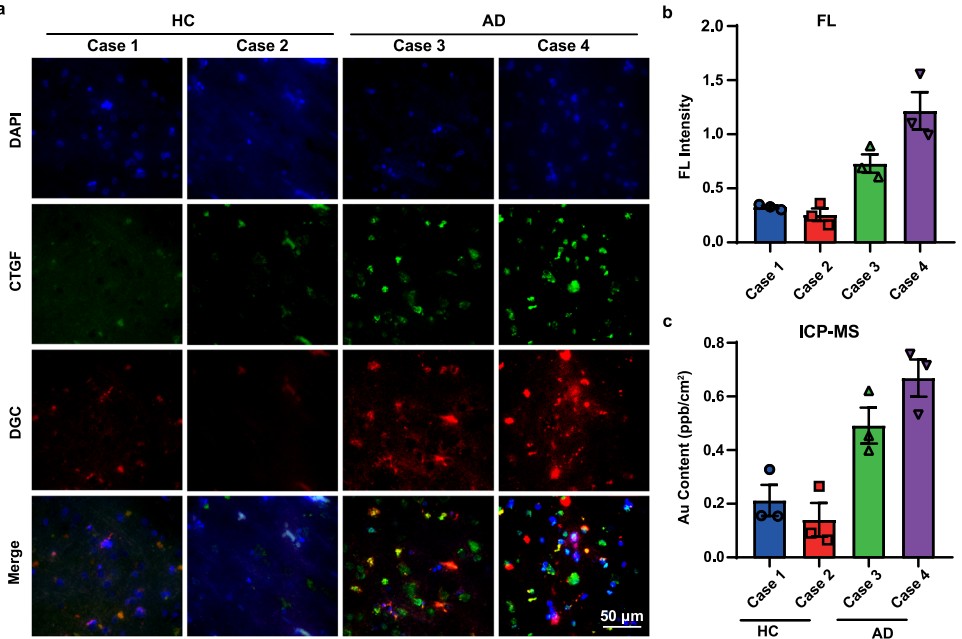

**Fig. 8 | DGC identifies early-stage AD patient brain sections. a** The fluorescence imaging of cortex regions in brain sections from an early stage of AD patient (AD) and Healthy control (HC) after staining with DGC (red fluorescence) and FITC-labeled CTGF antibody (green fluorescence). The merge pictures show the colocalization of DGC and CTGF. Scale bar = 50 µm. **b** The fluorescence (FL) intensity of DGC in (**a**) analyzed by ImageJ software. Data were presented as mean ± SD from three independent sections of the same case ($n = 3$). Student's $t$-test. **c** ICP-MS analysis of DGC (Au) content of per cm² after incubating DGC with indicated brain sections. Data were presented as mean ± SD from three independent sections of the same case ($n = 3$). Student's $t$-test. Source data are provided as a Source Data file.

culture medium. Next, the nuclei were stained with 5 µg/mL DAPI solution for 5 min. Finally, the cells were observed by a confocal laser scanning microscope (CLSM, PerkinElmer, USA). The fluorescence intensity of each cell was measured by ImageJ software, the red channel was extracted by clicking Split Channels, and the red fluorescent region, except the nucleus was selected by Freehand selections. Then the average fluorescence intensity in this region was measured and recorded for statistical analysis.

To further determine the specificity of DGC to CTGF, immunofluorescence staining was performed with CTGF antibodies to determine the location of CTGF in cells. First, U87MG cells in confocal dishes were fixed with 4% PFA for 30 min. Next, the cells were washed with PBS three times, and 3% bovine serum albumin (BSA) in PBS solution. Then, the cells were incubated with CTGF antibody (dilution 1:100, Abcam, Cat#ab6992) in PBS for 4 h at 4 °C, and FITC-labeled Goat Anti-Rabbit IgG (H + L) (1: dilution 1:200, Bioss, Cat#bs-0370R-FITC) in 3% BSA solution was added to the cells for another 30 min at 37 °C in the dark. Subsequently, 55 µM DGC were incubated with the cells for 1 h. Finally, the cells were stained with DAPI for 5 min and washed with PBS before the CLSM image (Nikon).

Moreover, a binding site blocking study was performed to confirm the specificity of the DGC to CTGF. First, U87MG cells were fixed with 4% PFA. Then, a PBS solution containing 3% BSA was used to block the nonspecific recognition sites for 30 min. After BSA blocking, the cells were pre-incubated with 5 mM of free DAG peptide for 1 h, followed by 55 µM DGC added for 60 min. Then, DAPI (5 µg/mL) was added to U87MG cells to stain the nuclei. Finally, the cells were imaged by the CLSM system.

**Visual observation of CTGF expression in cells by chromogenic DAB oxidation**

To observe accurately the expression level of CTGF in cells with the naked eye, we optimized the conditions for labeling U87MG, SH-SY5Y, and CTX TNA2 cells stained with DGC. According to the study, the optimal incubation concentration of DGC (40 µM) was used to determine the optimal culture conditions using the DAB Horseradish Peroxidase Color Development Kit. After the cells were co-stained by DGC at 4 °C overnight, a DAB chromogenic agent was added to the cells. Since DGC can label CTGF proteins on the cells, when DAB chromogenic is added, DGC on the cells can catalyze DAB to produce an insoluble brown precipitate to attach to the cells.

**Quantitating CTGF in cells by ICP-MS**

After U87MG, SH-SY5Y, and CTX TNA2 cells were imaged by a laser confocal microscopy system, these cells labeled with DGC were quantitatively analyzed by an ICP-MS system (Thermo Elemental X7, USA). Each cell group in the glass vials was digested by adding 5 mL aqua regia (HCl: $HNO_3$ = 3:1) overnight. Subsequently, the aqua regia was heated to evaporate to the last drop to be diluted by a mixture of 2% $HNO_3$ and 1% HCl. A standard curve of Au was made to quantify the concentration of Au in different cells. ICP-MS was used to test the Au signal in triplicates.

**Immunofluorescence analysis of CTGF in the brain sections of AD mice**

To evaluate the expression of CTGF in the brain of AD mice and the efficacy of DGC in vitro, brain sections of APP/PS1 transgenic mice (derived from C57BL/6J) from 1-month to 9-month-old (1-, 2-, 3-, 6- and 9-month-old, female, $n = 3$) were used as AD model, and brain sections of the age-matched wild-type C57BL/6J mice were used as controls (female, $n = 3$). The brain sections were subjected to CTGF antibody immunofluorescence staining and DGC fluorescence imaging. The brains were collected and fixed with 4% PFA. The embedded brain was then sectioned using a microtome into 5-µm thick consecutive sections. The brain sections were immersed in Tris-EDTA buffer (pH = 9.0) at 95 °C for 15 min. The brain sections were then incubated with 3% bovine serum albumin for 1.5 h. Then, the brain sections were stained overnight at 4 °C with an anti-CTGF primary antibody (dilution 1:50, Abcam, Cat#ab6992). After restoring room temperature, they were washed with TBST (pH = 7.4, 3 × 5 min) and incubated with FITC-

labeled Goat Anti-Rabbit IgG (h +L) (1: dilution 1:200, Bioss, Cat#bs-0370R-FITC) for 1 h. The brain sections were washed with TBST (pH = 7.4, 3 × 5 min). Finally, overnight staining with DGC (200 μM) at 4 °C and fluorescence imaging were obtained by the CLSM system.

## Multimodal analysis of CTGF in the brain sections of AD mice

Brain sections from female APP/PS1 double transgenic mice were used as AD models, and brain sections of the age-matched female wild-type C57BL/6J mice were used as controls to evaluate the availability of DGC in vitro. Three-month-old APP/PS1 mice (early-stage, unestablished Aβ plaques) and 9-month-old APP/PS1 mice (late-stage, established Aβ plaques) were chosen for comparison. All mice were purchased from Beijing HFK Bioscience Co., Ltd and handled in accordance with the Ethics Committee of Beijing University of Technology, China. To avoid the interference of strong background fluorescence signals of blood in cerebral microvessels, the blood vessels were emptied by cardiac perfusion with saline and PFA before dissociating the brain. After the completion of perfusion, the brain was collected and fixed with PFA for a longer time ex vivo. The embedded brain was then sectioned using a microtome into 5-μm thick continuous sections. The brain sections were labeled with DGC or CTGF, GFAP, and Aβ antibodies. Deparaffinization was performed first and antigen retrieval was implemented with immersion of the brain sections into Tris-EDTA buffer (pH = 9.0) at 95 °C for 15 min. Then, the brain sections were incubated with 3% bovine serum albumin for 1.5 h.

For fluorescence imaging of CTGF, the brain sections were stained with anti-CTGF primary antibody (dilution 1:50, Abcam, Cat#ab6992), anti-glial fibrillary acidic protein (GFAP) primary antibody (diluted 1:400, Proteintech, Cat#16825-1-AP) or anti-Aβ 17-24 (4G8, diluted 1:1000, BioLegend, Cat#SIG-39200) at 4 °C for 4 h. After they were washed by TBST (pH = 7.4, 3 × 5 min) at room temperature, the brain sections were incubated with the secondary antibody (FITC-labeled Goat Anti-Rabbit IgG (H + L) (1: dilution 1:200, Bioss, Cat#bs-0370R-FITC) for 1 h. After the brain sections were washed by TBST (pH = 7.4, 3 × 5 min). Finally, the brain sections were co-stained by DGC (200 μM) at 4 °C overnight, and the staining results were acquired on the CLSM system.

To observe the expression level of CTGF in the brain section with the naked eye, we optimized the conditions for labeling WT mice, 3-month-old APP/PS1 mice and 9-month-old APP/PS1 mice stained with DGC. According to the study, the optimal incubation concentration of DGC (200 μM) was used to determine the conditions using the DAB Horseradish Peroxidase Color Development Kit. After the brain sections were co-stained by DGC (200 μM) at 4 °C overnight, a DAB chromogenic agent was added to the brain sections. Since DGC possesses peroxidase-mimic activity, when DAB solution is added, DGC binding on the tissue can catalyze the oxidation of DAB to produce an insoluble brown precipitate in situ of the brain sections. After the brown product was imaged by optical microscopy and analyzed by Image Pro Plus software.

In situ visualization/quantification of CTGF in brain tissue by LA-ICP-TOF-MS. Brain tissue isolated from WT, 3-month-old APP/PS1 and 9-month-old APP/PS1 mice was collected after cardiac perfusion with pre-cooled normal saline. The embedded brain was sectioned to 5 μm using a microtome after freezing with liquid nitrogen. The frozen sections were taken out and placed at room temperature for 15 min, then soaked in water for 10 min, and washed with PBS. After incubation with 3% BSA solution for 60 min, the brain sections were stained with DGC (200 μM) overnight at 4 °C. On the second day, the DGC-labeled brain slices were washed with deionized water three times, and analyzed by LA-ICP-TOF-MS. An Iridia laser ablation system (Teledyne Photon Machines, Bozeman, USA) with 193 nm ArF excimer laser and low dispersion ablation unit was used. The LA system was coupled to icpTOF 2R ICP-TOF-MS (TOFWERK AG, Thun, Switzerland) by the Aerosol Rapid Introduction System (ARIS). Using high-purity helium as transport gas, the ablated aerosol samples were transported from the ablative chamber to ICP-MS. For LA-ICP-TOF-MS experiments, daily

tuning is performed using the NIST 612 glass standard RM (National Institute of Standards and Technology, Gaithersburg, USA) to obtain high sensitivity of $^{59}$Co, $^{115}$In, and $^{238}$U signals and low oxidation rates (e.g., UO$^+$/U$^+$ ≤3%). To obtain the shortest monopulse response, the flow of the internal and external cells is optimized. Laser spot size was set to 10 μm, line spacing was set to 10 μm, dosage was set to 1, laser frequency was set to 200 Hz. The parameters and conditions of the LA-ICP-TOF-MS are shown in Table S7. Imaging was performed using HDIP software (Teledyne Photon Machines, Bozeman, USA), TOFware, and Laser Image Viewer software (TOFWERK AG, Thun, Switzerland).

The DGC in the brain section was measured by ICP-MS. Briefly, the brain sections were digested with 5 mL mix solution (HNO$_3$ and H$_2$O$_2$ = 3:1) overnight. Then, each section was digested at a mild boiling temperature until the solution was evaporated to 0.1–0.2 mL and added 5 mL aqua regia (HCl: HNO$_3$ = 3:1) overnight. Then each section was digested at a mild boiling temperature until the solution was evaporated to 0.1– 0.2 mL and diluted by 2% HNO$_3$ and 1% HCl to the final volume.

## Parallel artificial membrane permeability assays (PAMPA)

A parallel artificial membrane permeability assay was performed in a 96-well precoated PAMPA plate system (Corning Gentest # 353015). The experimental solutions of DGC were prepared by diluting the stock solutions (10 mM) in PBS at a final concentration of 40 μM and then added to the donor portion of the plate (300 μL/well), while PBS (200 μL/well) was added to the acceptor portion, and the DMSO concentration was lower than 1%. The filter plate was then coupled with the receiver plate, and the plate assembly was incubated at room temperature without agitation for 5 h. Samples of the initial donor solution (C$_0$) were collected and stored at 4 ˚C. At the end of the incubation, samples were collected from the donor and acceptor plates and then added to aqua regia (HCl: HNO3, volume ratio = 3:1). The C$_0$ samples were treated similarly. The final concentrations of DGC in the donor, acceptor, and T0 wells were quantified by the ICP-MS system (Thermo Elemental X7, USA). The results were used to calculate an effective permeability (Pe, cm/s) value. Permeability of DGC was calculated using the following Eq. (1):

$$P_e = \{-ln[1\text{-}C_A(t)/C_{eq}]\}/[A \times (1/V_D + 1/V_A) \times t] \quad (1)$$

Where A = filter area (0.3 cm$^2$), $t$ = incubation time (seconds), C$_A$($t$) = compound concentration in acceptor well at time $t$, V$_D$ = donor well volume (0.3 mL), V$_A$ = acceptor well volume (0.2 mL), C$_D$($t$) = compound concentration in donor well at time $t$, and C$_{eq}$ was calculated using the following Eq. (2):

$$C_{eq} = [C_D(t) \times V_D + C_A(t) \times V_A]/(V_D + V_A) \quad (2)$$

## Monitoring the DGC in blood by ex vivo NIR-II imaging

The C57BL/6J mice (male and female) were injected with DGC (20 mg/kg) through the tail vein ($n$ = 3). After injection, the blood samples were collected at 0.083, 0.25, 0.5, 1, 2, 4, 8, 10, 12, 24, and 48 h through the posterior orbital vein of the mice. The serum was obtained by centrifugation (1200×$g$ for 5 min) to remove blood cells. The serum samples were observed through the In vivo Master (Grand-imaging Technology) equipped with an InGaAs camera C-RED2 (Firstlight) (λ ex = 808 nm, 1000 nm longpass (LP) filter (Edmund Optics)) camera to obtain the NIR-II image (exposure time: 50 ms). ImageJ software was used to process and analyze the images obtained in the experiment, and the DGC content in the sample was evaluated according to the fluorescence intensity of the sample.

## The pharmacokinetic study of DGC in C57BL/6J mice

Male and female mice were given a single intravenous dose of 20 mg/ kg ($n$ = 3). The blood of posterior orbital venous plexus was collected at

continuous time points (0.083, 0.25, 0.5, 1, 2, 4, 8, 10, 12, 24, and 48 h) and the serum was rapid separation by centrifugation (1200×$g$ at 4 °C, 10 min). Au content in the serum sample was detected by ICP-MS. The pharmacokinetic index was obtained by using winnonlin8.4 non-atrioventricular model.

### NIR-II imaging and multimodal analysis of CTGF in AD brain of APP/PS1 mice
All animal procedures were performed under an approved protocol of the Ethics Committee of Beijing University of Technology, China. Female APP/PS1 mice and age-matched wild-type C57BL/6J mice were shaved before background imaging. For NIR-II imaging in vivo with DGC, the mice ($n = 3$) were fasted overnight before the experiment and then injected intravenously with DGC (20 mg/kg, 200 μL). NIR-II imaging of the brains was carried out on an In vivo Master (Grand-imaging Technology) equipped with an InGaAs camera C-RED2 (Firstlight) at different time points after injection of DGC (λ ex = 808 nm, 1000 nm longpass (LP) filter (Edmund Optics)). The excitation intensity of the 808 nm laser (Wuhan Grand-imaging Technology) was about 100 mW/cm and the optical images were acquired using an exposure time of 50 ms. Imaging data were analyzed by ImageJ software.

Mice were then perfused intracardially with saline and 4% PFA, and major organs (including the brain, heart, liver, spleen, lung, and kidney were isolated. The dissociated brains were performed NIR-II imaging ex vivo using the same method as in vivo imaging ($n = 3$). All isolated organs were also imaged by the NIR-II imaging system (λ ex = 808 nm, 1000 nm longpass (LP) filter (Edmund Optics)). And then, the brains were kept in 4% PFA overnight and embedded with paraffin. Sections with 5-μm thickness were cut and analyzed by immunostaining with antibodies to CTGF (dilution 1: 50, Abcam, Cat#ab6992), GFAP (dilution 1:100, Proteintech, Cat#16825-1-AP), CD31 (dilution 1:200, Beyotime, Cat#AF6408), Neun (dilution 1:300, abcam, Cat# ab177487), and Iba 1 (dilution 1:50, Beyotime, Cat# AF7143), respectively. The colocalization of DGC with each antibody was observed in the brain sections by fluorescence microscope (Nikon).

To visualize the distribution of DGC-labeled CTGF in the brain, the brain sections were imaged by the LA-ICP-TOF-MS system. Brain tissue isolated from WT and 3-month-old APP/PS1 mice was collected, frozen with liquid nitrogen, and sectioned to 5 μm. The brain sections were stained with DGC and analyzed by LA-ICP-TOF-MS. Laser spot size was set to 20 μm, line spacing was set to 20 μm, dosage was set to 1, Laser frequency was set to 200 Hz. The parameters and conditions of the LA-ICP-TOF-MS are shown in Table S7. Imaging was performed using HDIP software, TOFware, and Laser Image Viewer software. Finally, The DGC in the brain section was measured by ICP-MS. Briefly, the brain sections were digested with 5 mL mix solution ($HNO_3$ and $H_2O_2$ = 3:1) overnight. Then, each section was digested at a mild boiling temperature until the solution was evaporated to 0.1– 0.2 mL and added 5 mL aqua regia (HCl: $HNO_3$ = 3:1) overnight. Then each section was digested at a mild boiling temperature until the solution was evaporated to 0.1– 0.2 mL and diluted by 2% $HNO_3$ and 1% HCl to the final volume.

### Multimodal analysis of CTGF in brain samples from AD patients
The brain tissue samples of autopsy-confirmed AD patients were obtained from the Chinese Brain Bank Center (CBBC) and Human Brain Bank, Central South University Xiangya School of Medicine. First, the brain sections were air-dried for 15 min at room temperature, followed by immersion in water. 5% $H_2O_2$ was applied for 15 min at room temperature. The sections were washed three times with PBS for 5 min each time. Then, tissue repair was performed using trypsin at a temperature of 37 °C for 5 min and washed three times with PBS. We chose 10% sheep serum to block for a duration of 20 min at 37°. Primary antibodies (CTGF antibody and Aβ antibody) were added to the sections overnight at 4 °C. FITC-labeled Goat Anti-Rabbit IgG (H + L) (1: dilution 1:200, Bioss, Cat#bs-0370R-FITC) was applied and incubated

at 37 °C for 20 min in darkness. Subsequent washing steps involved rinsing the sections three times with PBS, each time lasting for 5 min while avoiding exposure to light. DAPI staining was performed, taking care to store the samples away from light during the 5-min procedure. Then observed by the CLSM system. Then, we chose the optimal incubation concentration of DGC (200 μM) was used to determine the conditions using DAB Horseradish Peroxidase Color Development Kit. After the brain sections were co-stained by DGC (200 μM) at 4 °C overnight, a DAB chromogenic agent was added to the brain tissue sections to explore whether DGC can catalyze DAB solutions to produce insoluble brown precipitate to attach the brain tissue sections with an optical microscope. Finally, the brain tissue sections were digested with aqua regia to measure the Au content by the ICP-MS system.

### Statistical analysis
All statistical analysis were performed using GraphPad and Origin. The data were analyzed using Student's $t$-test, and represented as mean ± SD. *$p < 0.05$, **$p < 0.01$ as indicated in the figure legends. For each experiment, unless otherwise noted, the data under each condition were accumulated from at least three independent experiments.

### Reporting summary
Further information on research design is available in the Nature Portfolio Reporting Summary linked to this article.

## Data availability
The authors declare that all data supporting the findings of this study are available within the paper and its supplementary information files. Source data are provided with this paper.

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

## Acknowledgements

The authors also thank Dr. Lingna Zheng (Institute of High Energy Physics, Chinese Academy of Sciences) for the support with LA-ICP-TOF-MS detection, Prof. Wen Shi (Institute of Chemistry, Chinese Academy of Sciences) for the support with NIR-II imaging, Dr. Zi Yang (Protein Preparation and Characterization Core Facility of Tsinghua University Branch, China National Center for Protein Sciences) for the support with SPR analysis. We also thank the Chinese Brain Bank Center (CBBC, Wuhan, P. R. China) and Human Brain Bank, Central South University Xiangya School of Medicine for providing the brain tissues of autopsy-confirmed AD patients and normal controls. This work was supported by the National Key Research & Development Program of China (grant No. 2022YFA1207300, X.G. and 2021YFA1201000, Q.Y.), the National Natural Science Foundation of China (22334001, X.G.; U2067214, X.G.; 32171378, Q.Y.; and 22376009, K.C.).

## Author contributions

C.L. synthesized the probe and did cell and animal imaging, C.M. and Y.L. assisted animal studies, J.Y. assisted cell studies. X.R., L.G., D.S., K.C., and M.C. provided experimental suggestions. Q.Y. and X.G. designed the project and wrote the manuscript.

## Competing interests

The authors declare no competing interests.
