## [Peer Review File · Nature Communications]

Reviewers' Comments:

Reviewer #1:

Remarks to the Author:

In this manuscript, Lu et al. reported a novel gold cluster-based probe modified with peptides that specifically bind to connective tissue growth factor (CTGF). Owing to the elevated expression of CTGF is an upstream regulator of amyloid-beta (A β) plaque, the probe holds a promise for potential application in the early diagnosis of Alzheimer's disease (AD). The unique feature of this probe is their capacity to emit near-infrared (NIR) light, rendering them suitable for in vivo imaging. Moreover, their ability to target CTGF could potentially make them superior to probes that target A β plaques.

The article is well-crafted, and the experiments convincingly highlight the distinctive properties of the newly developed probe. I would suggest accepting it after the following concerns are addressed.

1. How long can the fluorescence last in the cells and AD brain, or how stable is the fluorescence of the probe?
2. How are DGC or DAG and CTGF dissociation equilibrium constants (KD) calculated and compared? If the protein of peptide DAG targets CTGF, what is the reason for the 1000-fold increase in the KD values after the formation of probe DGC?
3. In Fig 2c, the fluorescence intensity of U87 and SY5Y seems similar. Please clarify this and provide how to calculate the FI in the methods.
4. The ICP-MS of the whole brains from DGC-treated normal or APP/PS1 mice should be quantified, not only brain sections.
5. Authors only showed DGC staining in the cortex of mice brain, what about hippo? Is there any experiment or literature to support the content of CTGF at different positions in different periods?
6. Does the sensitivity of in vivo imaging in the NIR-II region differentiate between 3-month-old and 9-month-old AD mice?
7. In Fig 3b, in 3-month-old brain sections, the fluorescence signal of DGC overlaps well with the co-localization of antibody in slightly larger lesions, but does not overlap with small signals. Brain slices at 9 -month-old brain sections were slightly less effective. Why?
8. The authors emphasized the uniqueness of the NIR-II emission, I think it is necessary to add some discussions of the emission mechanism.

Reviewer #2:

Remarks to the Author:

The manuscript titled "A probe for NIR-II imaging and multimodal analysis of early Alzheimer's disease by targeting connective tissue growth factor" presents a novel peptide-coated gold nanocluster, referred to as DGC, which serves as a NIR-II emissive probe for detecting CTGF in the cortex of early AD mice and human brains. Authors well demonstrated the required analyses and biological experiments using the proposed materials. However, additional critical experiments are essential to substantiate the suitability of DGC nanoparticle for early diagnosis of AD, to be published in Nature Communications. Thus, I do not agree with acceptance of the current manuscript with the following reasons.

1. Authors need to provide direct evidence that the observed signal differences in the WT and APP/PS1 mouse brain result from varying CTGF concentrations, for example using ELISA. In suggested experimental condition, is the amount of CTGF in the AD model brain samples sufficiently higher than in the WT? Do these values correlate quantitatively or qualitatively with the fluorescence response obtained using DGC?
2. Authors explained that the developed DGC nanoparticle has high BBB penetrability. However, the lack of experimental results characterizing this BBB penetrability hinders the evaluation of the study's interpretations and main claims. Merely citing previous literatures is inadequate for a proper assessment of the experiments and their outcomes. Therefore, direct evidence providing

BBB penetrability, such as PAMPA assay is necessary. In this context, is there no difference between BBB penetrability in the in vivo imaging of WT and APP/PS1 mice? Typically, the BBB in AD model transgenic mouse is expected to be compromised.

3. In vivo stability and/or pharmacokinetic studies of DGC as a potential NIR-II imaging agents should be included in this manuscript. Additionally, as an initial study of CTGF measurement in early AD brain samples, authors should demonstrate results from narrow time points, not only 3- and 9-month-old, to provide robust evidence that CTGF-induced NIR signals from DGC are detectable at the early stages of AD.

4. In the human brain sample imaging results, portions stained by DGC are shown in unspecified extracellular areas. Authors should clarify why this observation contrasts with the results from cell experiments.

5. The manuscript briefly compares DGC with A β in Figure 5, showing significant co-localization in the areas of A β deposition. This could potentially interfere with accurate CTGF concentration measurements. Although the authors assert that the DGC assay is suitable for early AD diagnosis, the study only includes non-demented (healthy control) and advanced AD pathology patient samples, which do not represent early-onset disease. To substantiate claims of early diagnosis using patient samples, imaging studies should focus on stages where CTGF is expressed prior to A β accumulation.

6. The introduction is too brief to provide adequate context for the study. Authors should expand on the background information to explain the development of their strategy and its relevance to early AD diagnosis. Furthermore, the results and discussion sections are overly concise. A more detailed description of the finding and comprehensive discussion is required.

Reviewer #3:

Remarks to the Author:

In this work, the authors developed a CTGF-targeting peptide-coated gold nanocluster (DGC) that can emit both NIR-II and red fluorescence, allowing non-invasive NIR-II imaging of CTGF when there is no appearance of A β plaque deposition. Basically, the DGC probes were well designed and performed with a variety of characterizations. The authors also demonstrated the imaging potential of DGC probes in vitro and in vivo by some studies with analyzed data. The work has been solidly done, below is a list of technical issues for the authors to consider when revising their manuscript:

1. For the preparation of DGC probes, how to control the ratio of HAuCl₄ and DAG peptide?
2. The stability and quantum yield of DGC probes should be characterized.
3. The co-localization coefficient should be calculated in Figure 2d.
4. The scale bar should be provided in Figure 3a and Figure 4b. In addition, the semiquantitative analysis should be provided in Figure S7.
5. Since the authors claimed that the DGC probes can imaging CTGF in NIR-II window, what is the advantage of NIR-II imaging in this work? More data on NIR-II imaging should be provided.
6. The in vivo biosafety of DGC probes should be characterized by H&E staining and biochemical analysis.
7. How accurate or threshold can DGC be in detecting CTGF compared with currently reported probes?

REVIEWER COMMENTS

Reviewer #1 (Remarks to the Author):

In this manuscript, Lu et al. reported a novel gold cluster-based probe modified with peptides that specifically bind to connective tissue growth factor (CTGF). Owing to the elevated expression of CTGF is an upstream regulator of amyloid-beta (A β) plaque, the probe holds a promise for potential application in the early diagnosis of Alzheimer's disease (AD). The unique feature of this probe is their capacity to emit near-infrared (NIR) light, rendering them suitable for *in vivo* imaging. Moreover, their ability to target CTGF could potentially make them superior to probes that target A β plaques.

The article is well-crafted, and the experiments convincingly highlight the distinctive properties of the newly developed probe. I would suggest accepting it after the following concerns are addressed.

Reviewer 1: How long can the fluorescence last in the cells and AD brain, or how stable is the fluorescence of the probe?

Response: We monitored the fluorescence changes of DGC probes *in vitro*, in U87MG cells and in brain of APP/PS1 mice, respectively. Results showed that the fluorescence intensity of DGC decreased only slightly *in vitro* after two weeks storage (Fig R4). In cultured U87MG cells, DGC probe could be well internalized, and obvious fluorescence could still be observed in the cells after withdrawal of DGC-containing medium for 72 h (Fig R5). In APP/PS1 mice, we detected temporal changes of luminescence from DGC in the brain by *in vivo* NIR-II imaging. Results showed that the NIR-II fluorescence was still observed in the brains 24 h after injection (Fig R6). Considering that the elimination half-life ($t_{1/2z}$) of DGC in mice is about 20 h, we infer that DGC maintains a stable structure throughout the diagnostic testing time window. These results demonstrate that DGC has good stability *in vitro* and *in vivo*.

Figure R4. *in vitro* stability of DGC after two weeks storage in PBS.

Figure R5. Stability of DGC in U87MG cells.

Figure R6. Stability of DGC in AD brains. Monitoring fluorescence intensity of DGC in brain of WT mice and 3-month-old APP/PS1 mice for 24 h (left). *In vivo* and *ex vivo* NIR-II imaging of brains after DGC injected for 24 h (right).

Reviewer 1: How are DGC or DAG and CTGF dissociation equilibrium constants (KD) calculated and compared? If the protein of peptide DAG targets CTGF, what is the reason for the 1000-fold increase in the KD values after the formation of probe DGC?

Response: BIAcore 8K+ instrument (Pharmacia Biosensor AB) was used to detect the affinity between DGC probe or DAG peptide and recombinant human CTGF protein, and BIAcore Insight Evaluation 3.0 software was used for the real-time binding and kinetic analysis, as described in the METHODS section. The concentration of DGC was initially determined by ICP-MS quantification of Au. However, the affinity to CTGF should be compared as a complete molecule. Result of MALDI-TOF-MS indicated the precise molecular formula of DGC can be derived as Au₂₆DAG₈. Thus, **the accurate concentration of DGC molecules is obtained through dividing the Au concentration by 26.** The obtained KD of DAG and DGC was 2.23×10^{-5} M and 2.19×10^{-8} M, respectively, which is almost 1000 times. The much stronger affinity of DGC may be due to the multivalent (eight) display of DAG peptides on the surface of the probe, which is a well-established theory for enhancing target affinity (*J. Am. Chem. Soc.* 2020, 142, 4800-4806; *Bioconjugate Chem.* 2022, 33, 1922-1933; *Adv. Sci.* 2022, 9, 2103098; *ACS Chem. Biol.* 2023, 18, 1066-1075).

Reviewer 1: In Fig 2c, the fluorescence intensity of U87 and SY5Y seems similar. Please clarify this and provide how to calculate the FI in the methods.

Response: The analysis of fluorescence intensity in cells in Fig S2a is the average fluorescence intensity in each cell, rather than the fluorescence density at a single site, and the detailed analysis methods have been added to the revised methods section. The fluorescence intensity of each cell was measured by Image J software, the red channel was extracted by clicking Split Channels, and the red fluorescent region except the nucleus was selected by Freehand selections. Then the average fluorescence intensity in this region was measured and recorded for statistical analysis.

In Fig 2c, the fluorescence intensities of U87MG and SY5Y seems similar, possibly due to the different morphological sizes of the two cell lines. U87MG cell has a larger nucleus and cell body than SH-SY5Y cell, and also showed more extensive CTGF

expression, but the number of cells in the same field of view is significantly less than SH-SY5Y cell, maybe resulting in a similar perception of fluorescence intensity.

Reviewer 1: The ICP-MS of the whole brains from DGC-treated normal or APP/PS1 mice should be quantified, not only brain sections.

Response: According to the suggestion, the whole brain samples collected after *ex vivo* NIR-II imaging in Fig. 4 were frozen grinded and digested, and the content of Au in the whole brain was measured by ICP-MS. The results showed that the content of Au in the brain of APP/PS1 mice was more than twice that of the age-matched WT mice (Fig R7), which was consistent with the results of the NIR-II imaging.

Figure R7. The Au content in brains of DGC-injected 3-month-old APP/PS1 mice and the age-matched WT mice (n = 3).

Reviewer 1: Authors only showed DGC staining in the cortex of mice brain, what about hippo? Is there any experiment or literature to support the content of CTGF at different positions in different periods?

Response: According to previous reports, the expression pattern of CTGF was slightly different in several AD mouse models. The homing of DAG peptide was mainly seen in the cortex of the Tg2576 mice and in the hippocampus of the hAPP-J20 mice (*Nat. Commun.* 2017, 8, 1403). A marked increase of CTGF levels in the brains of APP/PS1 transgenic mice were also observed, and is mainly located in the cortical (*Mol. Neurobiol.* 2012, 45, 440-454; *Hum. Mol. Genet.* 2017, 26, 3909-3921). In this study, DGC signals were observed in both the hippocampus and the cortex of 9-month-old

APP/PS1 mice, and showed more obvious in the cortex than the hippocampus, as shown in Figure 3a. However, in the brains of 3-month-old APP/PS1 mice, we only observed significant DGC signals in the cortex, but almost none in the hippocampus, as shown in below (Fig R8, hippocampus with DAB color). Therefore, we only showed DGC staining in the cortex in Figure 3b.

Figure R8. Enlarged hippocampus of DAB chromogenic images of WT mice (left), 3-month-old APP/PS1 mice (middle) and 9-month-old APP/PS1 mice (right).

Reviewer 1: Does the sensitivity of *in vivo* imaging in the NIR-II region differentiate between 3-month-old and 9-month-old AD mice?

Response: To answer this question, we performed *in vivo* and *ex vivo* NIR-II imaging of 3-month-old and 9-month-old APP/PS1 mice after intravenous injection of DGC, simultaneously. Result showed that the overall NIR-II fluorescence intensity in brain of 9-month-old APP/PS1 mice is obvious higher than that of 3-month-old mice, both *in vivo* or *ex vivo* (Fig. R9), indicating that DGC probe has the ability and sensitivity to distinguish between early and late-stage AD mice.

Figure R9. The *in vivo* NIR-II images of WT mice (left), 3-month-old APP/PS1 mice (middle) and 9-month-old APP/PS1 mice (right) after intravenous injection of DGC for 8 h. And the *ex vivo* imaging of brains isolated from the mice (cardiac perfusion with saline, $\lambda_{ex} = 808$ nm).

Reviewer 1: In Fig 3b, in 3-month-old brain sections, the fluorescence signal of DGC overlaps well with the co-localization of antibody in slightly larger lesions, but does not overlap with small signals. Brain slices at 9-month-old brain sections were slightly less effective. Why?

Response: We speculate that large lesions may be more easily labeled by both antibodies and probes, so there will be better co-localization. From the overall fluorescence pattern, it can be found that the 9-month-old brain sections have more extensive CTGF expression than the 3-month-old brain sections, and the labeling efficiency of antibody and DGC probe may be affected, that resulting in slightly less effective of co-localization in small lesions. However, it was clear that CTGF expression was higher in the brain of 9-month-old AD mice than 3-month-old ones.

Reviewer 1: The authors emphasized the uniqueness of the NIR-II emission, I think it is necessary to add some discussions of the emission mechanism.

Response: Understanding the exact origins of NIR-II photoluminescence (PL) of gold nanoclusters (AuNCs) is of great significance for the development of highly effective NIR-II probes. However, the mechanism is very complicated and still under debate (*Chem. Soc. Rev.* 2019, 48, 2422–2457). So far, only Au₂₅ clusters have been

systematically studied for the correlation of PL origins to their structures, through experimental characterizations and theoretical calculations (*J. Phys. Chem. Lett.* 2021, 12, 1514–1519). By examining different surface ligands, it was shown that the NIR emission of Au₂₅ cluster was likely originated from the Au₁₃ icosahedral core state (*J. Phys. Chem. Lett.* 2021, 12, 1514–1519). It was found that the NIR-II PL of AuNCs can be enhanced by increasing the rigidity of surface ligands (*J. Am. Chem. Soc.* 2015, 137, 8244–8250), and the intrinsic structural rigidity also strongly affect the NIR PL quantum efficiency of thiolated AuNCs (*J. Am. Chem. Soc.* 2020, 142, 12140–12145). Recent studies indicated the great significance of the central Au atom and the surface “lock rings” as well the surface “lock atoms” in enhancing the NIR-II PL of AuNCs by suppressing the nonradiative decay (*ACS Nano.* 2021, 15, 16095–16105; *Small.* 2021, 17(11), 7). In summary, the NIR-II PL emission of AuNCs is related to the Au core and the surface ligands as well the charge transfer states, which leads to the complexity of their fluorescence origin (*Coord. Chem. Rev.* 2021, 448, 214184).

Many studies have focused on enhancing and modulating the NIR-II PL of AuNCs, but this remains a challenging task. Currently, metal atom doping (such as Cd, Cu, Zn, Pt and Ag) and ligand engineering (such as ligand exchange reactions, ligand properties regulation and surface rigidification) have been shown to be two major promising strategies for enhancing the NIR-II PL of AuNCs (*Sci. Adv.*, 2023, 9, eadh7828; *J. Am. Chem. Soc.*, 2023, 145, 26328-26338; *ACS Appl. Nano Mater.*, 2023, 6(17), 15945-15958; *Adv. Mater.*, 2019, 1901015). In particular, the NIR-II PL brightness and quantum yield (QY) of AuNCs can be significantly improved by single metal atom doping, for example, Au₂₅Cd₁NCs exhibited about 56 times higher NIR-II PL intensity than Au₂₅NCs, which may be the direction of further optimization of DGC probes (*Small.*, 2023, 19(30), 2300145).

Reviewer #2 (Remarks to the Author):

The manuscript titled “A probe for NIR-II imaging and multimodal analysis of early Alzheimer's disease by targeting connective tissue growth factor” presents a novel peptide-coated gold nanocluster, referred to as DGC, which serves as a NIR-II emissive probe for detecting CTGF in the cortex of early AD mice and human brains. Authors well demonstrated the required analyses and biological experiments using the proposed materials. However, additional critical experiments are essential to substantiate the suitability of DGC nanoparticle for early diagnosis of AD, to be published in Nature Communications. Thus, I do not agree with acceptance of the current manuscript with the following reasons.

Reviewer 2: Authors need to provide direct evidence that the observed signal differences in the WT and APP/PS1 mouse brain result from varying CTGF concentrations, for example using ELISA. In suggested experimental condition, is the amount of CTGF in the AD model brain samples sufficiently higher than in the WT? Do these values correlate quantitatively or qualitatively with the fluorescence response obtained using DGC?

Response: Thanks for the valuable suggestion. According to previous reports, the elevated CTGF in the brain of APP/PS1 mice was mainly secreted by the reactive astrocytes and was located primarily in the cortex rather than uniformly distributed throughout the brain (*Mol. Neurobiol.* 2012, 45, 440-454; *Hum. Mol. Genet.* 2017, 26, 3909-3921). In order to more accurately characterize the expression pattern in AD brain, we used immunofluorescence staining to detect CTGF expression in the brain sections of APP/PS1 mice from 1-month to 9-month-old according to the editor's and Reviewer's suggestion, and the age-matched WT mice were used as controls. The results of immunofluorescence staining of CTGF antibody showed that the expression level of CTGF in the cortex of APP/PS1 mice gradually increased with the increase of age, with a significant linear correlation, while the expression level in the brain of age-matched WT mice was significantly lower (Figure R1).

Figure R1. Immunofluorescence staining of CTGF and DGC probe on APP/PS1 mouse brain sections and the age-matched wild-type (WT) brain sections from 1- to 9-month-old.

To evaluate whether the amount of CTGF in the AD brain samples sufficiently higher than that in age-matched WT brain in the same experimental condition, we compared the expression level of CTGF in the brain lysates of early-stage (1- and 2-month-old) mice by ELISA. The ELISA quantitative results showed that the expression of CTGF in APP/PS1 mice brain was significantly higher than that of WT mice at the same age (Fig. R10).

Figure R10. ELISA quantification of CTGF in brain lysates of early-stage APP/PS1 mice (1- and 2-month-old) and the age-matched wild-type (WT) brains (n = 3).

To accurately evaluate the correlation between DGC probe and CTGF expression, DGC was simultaneously labeled on the brain sections that immunofluorescence stained by CTGF antibody (Fig. R1). Results showed that DGC and CTGF antibody were well co-localized in the brain sections, and there was a linear correlation with increasing CTGF expression. In addition, we performed *ex vivo* NIR-II imaging and luminescence semi-quantification of the brains of APP/PS1 mice aged from 1- to 9-month after DGC probe injection. Results showed that with the increase of age of APP/PS1 mice, the NIR-II luminescence intensity of DGC in the brain became stronger and stronger, and the semi-quantitative results showed a linear correlation (Fig. R11), which was consistent with the results of immunofluorescence analysis of brain slices.

Figure R11. *Ex vivo* NIR-II imaging and luminescence semi-quantification of the brains of APP/PS1 mice aged from 1- to 9-month (n = 3).

Reviewer 2: Authors explained that the developed DGC nanoparticle has high BBB penetrability. However, the lack of experimental results characterizing this BBB penetrability hinders the evaluation of the study's interpretations and main claims. Merely citing previous literatures is inadequate for a proper assessment of the experiments and their outcomes. Therefore, direct evidence providing BBB penetrability, such as PAMPA assay is necessary. In this context, is there no difference between BBB penetrability in the *in vivo* imaging of WT and APP/PS1 mice? Typically, the BBB in AD model transgenic mouse is expected to be compromised.

Response: According to the suggestion, PAMPA assay was performed and the test concentration of DGC was determined by reference to the blood concentration. Results showed that the permeability Mean P_e of DGC is higher than 1.5×10^{-6} cm/s at several concentrations, and the P_e value of 40 μ M DGC is $2.15 \pm 0.44 \times 10^{-6}$ cm/s, indicating a high BBB permeability (Table R2).

Table R2. The effective permeability coefficient of DGC

Sample	PAMPA permeability Mean P_e (10^{-6} cm/s)
DGC	2.15 ± 0.44 (n = 3)

On the other hand, before the *ex vivo* NIR-II imaging of AD brains, DGC probes that did not cross BBB were removed by cardiac perfusion with saline. Therefore, the

significant NIR-II luminescence within the brains come from the probes that have already crossed the BBB, demonstrating that DGC can effectively penetrate the BBB. Indeed, studies have shown that when the disease is fully developed (9 months of age) in AD mice, the permeability of the BBB increases, that perhaps increasing DGC leakage. However, there was no significant difference in BBB permeability between AD and WT mice in early stage (*Nat. Commun.*, 2017, 8, 1403). Therefore, we compared the brains of APP/PS1 mice from 1- to 3- months of age with the age-matched WT mice using *ex vivo* NIR-II imaging. Results showed that AD brains exhibited significantly stronger DGC NIR-II luminescence than that of age-matched WT brains, as early as 1 month of age, when BBB had not yet been significantly leaked in this context (Fig. R2). These results suggest that the increase of DGC in AD brains is not caused by BBB leakage, but is related to CTGF expression.

Figure R2. *Ex vivo* NIR-II imaging and fluorescence semi-quantitative of brains from 1-, 2-, 3-month-old APP/PS1 mouse and the age-matched WT mouse after cardiac perfusion with saline (n = 3).

Reviewer 2: In vivo stability and/or pharmacokinetic studies of DGC as a potential NIR-II imaging agents should be included in this manuscript. Additionally, as an initial study of CTGF measurement in early AD brain samples, authors should demonstrate results from narrow time points, not only 3- and 9-month-old, to provide robust evidence that CTGF-induced NIR signals from DGC are detectable at the early stages of AD.

Response: The *in vivo* stability of DGC was evaluated by *in vivo* NIR-II imaging of AD brains and serum fluorescence monitoring. Results showed that NIR-II

fluorescence was still observed in the brains 24 h after injection, which indicated a stable structure (Fig R6). Considering that DGC has an elimination half-life ($t_{1/2z}$) of about 20 h in mice, we infer that DGC maintains a stable structure throughout the diagnostic testing time window.

Figure R6. Stability of DGC in AD brains. Monitoring fluorescence intensity of DGC in brain of WT mice and 3-month-old APP/PS1 mice for 24 h (left). *In vivo* and *ex vivo* NIR-II imaging of brains after DGC injected for 24 h (right).

DGC in serum was quantitatively monitored by both NIR-II imaging and ICP-MS. Results showed that DGC emitted obvious NIR-II fluorescence in serum, and the change of fluorescence intensity was consistent with the quantitative results of ICP-MS, indicating that DGC existed in blood as an unbroken structure and had good stability *in vivo* (Fig. R12).

Figure R12. Serum fluorescence monitoring and ICP-MS quantification of DGC in serum after tail vein injection.

The pharmacokinetic study of DGC was performed according to the request. Detailed parameters are attached below (Table R3). Results showed that the elimination half-life ($t_{1/2z}$) of DGC in male and female mice was 21.38 h and 19.80 h, respectively, indicating a longer metabolic time than small molecule probes, which is more suitable for *in vivo* diagnostic applications.

Table R3. Values of parameters of the Pharmacokinetic of DGC in C57/6J mice.

Parameter	Unit	Value	
		Female (n=3)	Male (n=3)
AUC(0-t)	h*mg/L	651.00	615.22
AUC(0-∞)	h*mg/L	779.37	735.65
A μ MC(0-t)	h*h*mg/L	10136.52	8803.77
A μ MC(0-∞)	h*h*mg/L	19964.46	18299.37
MRT(0-t)	h	15.57	14.31
MRT(0-∞)	h	25.62	24.88
$t_{1/2z}$	h	19.80	21.38
Tmax	h	0.08	0.08
CLz/F	L/h/kg	0.03	0.03
Vz/F	L/kg	0.73	0.84
Cmax	mg/L	191.26	172.69

To demonstrate the detectable of DGC in brains at the early stages of AD, APP/PS1 mouse in more narrower time points (1-, 2- and 3-month-old) were detected by *in vivo* and *ex vivo* NIR-II imaging, and the age-matched WT mouse were used as controls. *In vivo* imaging results showed that the brain fluorescence intensity of AD mice was higher than that of age-matched WT mice (Fig. R13). Considering that black hair on the body surface and the residual probe in cerebrovascular may affect the accuracy of *in vivo* imaging, *ex vivo* analysis of brains after cardiac perfusion with saline can provide more reliable evidence. The *ex vivo* NIR-II imaging clearly showed that DGC fluorescence

in AD brain was significantly enhanced compared with that in age-matched WT brains, and the difference was significant even at 1 month of age (Fig. R2).

Figure R13. The *in vivo* NIR-II images of APP/PS1 mice (right) and age-matched WT mice (left) from 1-month to 3-month-old after intravenous injection of DGC.

Figure R2. *Ex vivo* NIR-II imaging and fluorescence semi-quantitative of brains from 1-, 2-, 3-month-old APP/PS1 mouse and the age-matched WT mouse after cardiac perfusion with saline (n = 3).

Reviewer 2: In the human brain sample imaging results, portions stained by DGC are shown in unspecified extracellular areas. Authors should clarify why this observation contrasts with the results from cell experiments.

Response: CTGF is a secretory protein that overexpressed by reactive astrocytes in the context of AD, which will be secreted to the extracellular region proximity to reactive astrocytes. Therefore, DGC staining in extracellular areas can be observed in the brain section, in addition to astrocyte cell bodies (*Nat. Commun.*, 2017, 8, 1403; *BMC Neurosci.*, 2003, 116(1), 1-6). However, staining in cells may lose extracellular

secretory portions due to cell fixation and rinsing steps, similar results can be found in previous reported work (*Nat. Commun.*, 2017, 8, 1403).

FAM-DAG labeled human brain sections of AD patient (left) and hiPSC-derived BMECs cells (right) from *Nat. Commun.*, 2017, 8, 1403

Reviewer 2: The manuscript briefly compares DGC with A β in Figure 5, showing significant co-localization in the areas of A β deposition. This could potentially interfere with accurate CTGF concentration measurements. Although the authors assert that the DGC assay is suitable for early AD diagnosis, the study only includes non-demented (healthy control) and advanced AD pathology patient samples, which do not represent early-onset disease. To substantiate claims of early diagnosis using patient samples, imaging studies should focus on stages where CTGF is expressed prior to A β accumulation.

Response: According to the suggestion, we obtained another two brain slices of AD patients at a very early stage according to the Thal standard and two healthy control samples from the Department of Neurology, Xiangya Hospital, Central South University (Table R1). The two early AD brain slices only observed very little A β accumulation in the neocortex and were therefore clinically identified to be Thal I stage by Xiangya Hospital, Central South University (*Neurology*. 2002, 58, 1791-1800; *Mol. Neurodegener.* 2019, 14, 32). What's exciting is that the fluorescence signal of DGC

probe and CTGF antibody in these two brain slices was strong and significantly different from that of the healthy controls. The quantitative results of ICP-MS also verified the significant difference. Moreover, the DGC probe (red) can co-locate well with CTGF immunofluorescence staining (green) on the AD brain slices (Fig. R3). These results suggest that DGC probe can identify AD brain prior to obvious A β accumulation, with a potential for early diagnosis of AD patient.

Table R1. The detailed information of brain tissue donors

ID Number	Age	Gender	Pathological diagnosis	A β staging (Thal)
Case 1	76	M	Health control (HC)	0
Case 2	70	F	Health control (HC)	0
Case 3	95	M	AD	1
Case 4	90	M	AD	1

Figure R3. (a) The fluorescence imaging of cortex regions in brain slices from early stage of AD patient (without obvious A β accumulation) and Healthy control (HC) after staining with DGC (red fluorescence) and FITC labeled-CTGF antibody (green fluorescence). The merge pictures show the colocalization of DGC and CTGF. Scale bar = 50 μ m. (b) The fluorescence intensity of DGC in

Figure (a) analyzed by Image J software. (c) ICP-MS analysis of DGC (Au) content of per cm² after incubating DGC with indicated brain slices.

Reviewer 2: The introduction is too brief to provide adequate context for the study. Authors should expand on the background information to explain the development of their strategy and its relevance to early AD diagnosis. Furthermore, the results and discussion sections are overly concise. A more detailed description of the finding and comprehensive discussion is required.

Response: Thanks for the suggestion, we have expanded the Introduction and Discussion sections in the revised manuscript. We mainly summarized and discussed the development and significance of CTGF in the early diagnosis of AD, as well as the luminescence mechanism and optimization direction of NIR-II emitting gold nanocluster probes for biomedical applications.

Reviewer #3 (Remarks to the Author):

In this work, the authors developed a CTGF-targeting peptide-coated gold nanocluster (DGC) that can emit both NIR-II and red fluorescence, allowing non-invasive NIR-II imaging of CTGF when there is no appearance of A β plaque deposition. Basically, the DGC probes were well designed and performed with a variety of characterizations. The authors also demonstrated the imaging potential of DGC probes *in vitro* and *in vivo* by some studies with analyzed data. The work has been solidly done, below is a list of technical issues for the authors to consider when revising their manuscript:

Reviewer 3: For the preparation of DGC probes, how to control the ratio of HAuCl₄ and DAG peptide?

Response: To establish and optimize of the synthesis procedure of DGC, we explored the ratio of precursor and reaction condition based on the ratio of Au atom to DAG peptide, and determined the optimal ratio and reaction kinetic parameters by detecting the fluorescence intensity of the product (Fig. R14). The optimal ratio of Au atom to DAG peptide was finally determined to be 1:1 in this study.

Figure R14. Optimization of the synthesis procedure of DGC by regulating precursor ratio, reaction temperature and reaction temperature.

Reviewer 3: The stability and quantum yield of DGC probes should be characterized.

Response: The stability of DGC *in vitro*, in cells and in APP/PS1 mice were determined respectively. Results showed that the fluorescence intensity of DGC decreased only slightly *in vitro* for two weeks storage (Fig R4). In cultured U87MG cells, DGC probe could be well internalized, and obvious fluorescence could still be observed in the cells after 72h incubation (Fig R5). In APP/PS1 mice, we detected temporal changes of DGC

in the brain by in vivo NIR-II imaging. Results showed that NIR-II fluorescence was still observed in the brains 24h after injection (Fig R6). Considering that DGC has a half-life of about 20h in mice, we infer that DGC maintains a stable structure throughout the diagnostic testing time window. These results demonstrate that DGC has good stability for potential AD diagnostic applications.

Figure R4. *In vitro* stability of DGC after two weeks storage in PBS.

Figure R5. Stability of DGC in U87MG cells.

Figure R6. Stability of DGC in AD brains. Monitoring fluorescence intensity of DGC in brain of WT mice and 3-month-old APP/PS1 mice for 24 h (left). *In vivo* and *ex vivo* NIR-II imaging of brains after DGC injected for 24 h (right).

The absolute quantum yield (QY) of DGC probe was measured to be $\sim 0.7\%$ using an integrated sphere technique, which was comparable to other reported gold nanoclusters for biological applications. Although this QY is not high, but sufficient to produce a bright signal in the AD mouse brain for detection *in vivo* in this study.

Reviewer 3: The co-localization coefficient should be calculated in Figure 2d.

Response: Thanks for the suggestion, and the co-localization coefficient in Figure 2d was calculated by Image J software. The co-localization coefficient (Pearson's correlation coefficient) of DGC and CTGF antibodies in Figure 2d was 0.946 (Fig. R15).

Figure R15. The co-localization coefficient (Pearson's correlation coefficient) of DGC and CTGF antibody in U87MG cells.

Reviewer 3: The scale bar should be provided in Figure 3a and Figure 4b. In addition, the semiquantitative analysis should be provided in Figure S7.

Response: We are sorry for the missing scale bar in Figure 3a and Figure 4b. We have added the correct scale bar in the revised Figure 3a and Figure 4b.

Revised Figure 3a after adding the scale bar

Revised Figure 4b after adding the scale bar

The Figure S7 was detected by red fluorescence (λ_{ex} = 500 nm, λ_{em} = 600 nm), which may be affected by the biological background. Therefore, in order to more accurately evaluate the biodistribution of DGC probes *in vivo*, we performed NIR-II imaging

($\lambda_{\text{ex}} = 808 \text{ nm}$) and semi-quantitative analysis, and the results have been replaced in the revised supplementary material (Revised Fig. S7).

Revised Figure S7. Biodistribution of DGC in APP/PS1 mice and age-matched WT mice detected by NIR-II imaging (n = 3).

Reviewer 3: Since the authors claimed that the DGC probes can imaging CTGF in NIR-II window, what is the advantage of NIR-II imaging in this work? More data on NIR-II imaging should be provided.

Response: NIR-II probes for biological imaging have some common advantages, such as noninvasive, high signal-to-noise ratio, deeper penetration depth, and elevated imaging resolution owing to reduced photon scattering, light absorption and autofluorescence. In this study, DGC probe showed high specificity and sensitivity for CTGF NIR-II imaging *in vivo* and *ex vivo*. Although the QY of DGC is not very high, but sufficient to produce a bright signal in the AD mouse brain for detection *in vivo* with high sensitivity. To demonstrate the detectable sensitivity of DGC in brains at the early stages of AD, APP/PS1 mouse in more narrower time points (1-, 2- and 3-month-old) were detected by *in vivo* and *ex vivo* NIR-II imaging. Results showed that with the increase of age of APP/PS1 mice, the NIR-II luminescence intensity of DGC in the brain became stronger and stronger (Fig. R16). Moreover, the *ex vivo* NIR-II imaging clearly showed that DGC fluorescence in AD brain was significantly enhanced compared with the age-matched WT brains, even at 1-month-old mice, demonstrating a high sensitivity of DGC on AD diagnosis (Fig. R2).

Figure R16. The *in vivo* NIR-II images of APP/PS1 mice from 1-month to 3-month-old after intravenous injection of DGC.

Figure R2. *Ex vivo* NIR-II imaging and fluorescence semi-quantitative of brains from 1-, 2-, 3-month-old APP/PS1 mouse and the age-matched WT mouse after cardiac perfusion with saline (n = 3).

Reviewer 3: The *in vivo* biosafety of DGC probes should be characterized by H&E staining and biochemical analysis.

Response: According to the request, the biosafety of DGC in C57/6J mice was evaluated by blood tests (blood cells and biochemical analysis) and organ pathological analysis (H&E staining). Results indicated that 20 mg/kg DGC did not induce any abnormal in blood parameters and major organs, compared with the untreated normal mice, indicating good biosafety for *in vivo* diagnostic applications (Table R4 and R5, Fig. R17).

Table R4. Effect of DGC on hematology of mice

Parameter	Short name	Ctrl results (n=3)	DGC results (n=3)	Unit	Normal parameter range
White blood cell count	WBC	7.12 ± 2.72	8.03 ± 1.12	10 ⁹ /L	0.80 - 10.60
Neutrophil count	Neu#	1.08 ± 0.6	1.07 ± 0.24	10 ⁹ /L	0.23 - 3.60
Lymphocyte count	Lym#	5.74 ± 1.84	6.65 ± 0.78	10 ⁹ /L	0.60 - 8.90
Monocytes count	Mon#	0.23 ± 0.26	0.25 ± 0.1	10 ⁹ /L	0.04 - 1.40
Eosinophil granulocyte count	Eos#	0.05 ± 0.04	0.04 ± 0.02	10 ⁹ /L	0.00 - 0.51
Basophil granulocyte count	Bas#	0.01 ± 0.01	0.02 ± 0.01	10 ⁹ /L	0.00 - 0.12
Percentage of neutrophil	Neu%	14.47 ± 3.87	13.2 ± 1.31	%	6.5 - 50.0
Percentage of lymphocyte	Lym%	81.93 ± 5.85	83 ± 2.08	%	40.0 - 92.0
Percentage of monocytes	Mon%	2.7 ± 2.23	3.07 ± 0.84	%	0.9 - 18.0
Percentage of eosinophil granulocyte	Eos%	0.67 ± 0.31	0.47 ± 0.23	%	0.0 - 7.5
Percentage of Basophil granulocyte	Bas%	0.23 ± 0.12	0.27 ± 0.06	%	0.0 - 1.5
Red blood cell count	RBC	10.45 ± 0.27	10.16 ± 0.42	10 ¹² /L	6.50 - 11.50
Hemoglobin	HGB	166.67 ± 5.51	161.67 ± 8.02	g/L	110 - 165
Hematocrit	HCT	52 ± 0.87	51.2 ± 1.93	%	35.0 - 55.0
Mean corpuscular volume	MCV	49.77 ± 0.45	50.4 ± 0.17	fL	41.0 - 55.0
Mean hemoglobin content of red blood cells	MCH	15.93 ± 0.31	15.93 ± 0.25	pg	13.0 - 18.0
Mean concentration of red blood cell hemoglobin	MCHC	320 ± 8.19	315.67 ± 5.69	g/L	300 - 360
Coefficient of variation of red blood cell distribution width	RDW-CV	13.7 ± 0.3	13.13 ± 1	%	12.0 - 19.0
Standard deviation of red blood cell distribution width	RDW-SD	28.77 ± 0.95	27.8 ± 2.1	fL	23.0 - 39.0
Platelet count	PLT	1359 ± 102.83	1228.67 ± 179.01	10 ⁹ /L	400 - 1600

Mean platelet volume	MPV	5.53 ± 0.15	5.53 ± 0.15	fL	4.0 - 6.2
Width of platelet distribution	PDW	15.47 ± 0.06	15.23 ± 0.25		12.0 - 17.5
Thrombocytopenia	PCT	0.75 ± 0.04	0.68 ± 0.09	%	0.100 - 0.780

Table R5. Effect of DGC on blood Biochemistry of mice

Parameter	Short name	Ctrl results (n=3)	DGC results (n=3)	Unit	Normal parameter range
Alanine aminotransferase	ALT	33.87 ± 1.87	32.8 ± 4.26	U/L	10.06-96.47
Aspartate aminotransferase	AST	150.90 ± 44.96	120.2 ± 20.04	U/L	36.31-235.48
Alkaline phosphatase	ALP	206.00 ± 89.66	220.7 ± 27.07	U/L	22.52-474.35
Urea	UREA	11.16 ± 0.92	9.62 ± 1.25	mmol/L	10.81-34.74
Creatinine	CREA	22.87 ± 2.79	22.67 ± 1.98	μmol/L	10.91-85.09

Figure R17. Effect of DGC on histology of major organs in mice.

Reviewer 3: How accurate or threshold can DGC be in detecting CTGF compared with currently reported probes?

Response: To our best knowledge, there was no CTGF targeting NIR-II probe has been reported. Previously, CTGF in brain sample was detected *in vitro* by antibodies, FAM-labeled DAG peptides and DAG-conjugated silver nanoparticles (*Nat. Commun.*, 2017, 8, 1403; *BMC Neurosci.*, 2003, 116(1), 1-6; *Hum. Mol. Genet.*, 2017, 26, 3909-3921). However, these tools can only be used for *in vitro* detection of cells and brain slices, they cannot be used for imaging and analysis *in vivo*. In this study, the affinity of DGC

probe with CTGF is about 1000-folds higher than that of free DAG peptide, and it can distinguish AD mice brain from the age-matched wild-type brain in 1-month-old mice (Fig. R18), exhibited a high sensitivity for diagnostic application.

Figure R18. DGC can distinguish 1-month-old AD mice from the age-matched wild-type mice.

Reviewers' Comments:

Reviewer #1:

Remarks to the Author:

The author has addressed all the issues. I recommend this manuscript for publication in Nature Communications.

Reviewer #2:

Remarks to the Author:

The authors have fully addressed reviewers' comments and the revised version is worth being accepted for publication as it is.

Reviewer #3:

Remarks to the Author:

The manuscript has been carefully revised according to the comments and suggestions raised by the reviewers. The overall language of the manuscript is well written in this revised manuscript. The Data has been well represented and interpreted. Therefore, I would like to recommend this manuscript to be published in Nature Communications.